# Universal Link Predictor By In-Context Learning on Graphs

**Kaiwen Dong**                                                                      *kdong2@nd.edu*
*University of Notre Dame*

**Haitao Mao**                                                                       *haitaoma@msu.edu*
*Michigan State University*

**Zhichun Guo**                                                                      *zguo5@nd.edu*
*University of Notre Dame*

**Nitesh V. Chawla**                                                                 *nchawla@nd.edu*
*University of Notre Dame*

**Reviewed on OpenReview:** *https://openreview.net/forum?id=EYpqmoejB8*

## Abstract

Link prediction is a crucial task in graph machine learning, where the goal is to infer missing or future links within a graph. Traditional approaches leverage heuristic methods based on widely observed connectivity patterns, offering broad applicability and generalizability without the need for model training. Despite their utility, these methods are limited by their reliance on human-derived heuristics and lack the adaptability of data-driven approaches. Conversely, parametric link predictors excel in automatically learning the connectivity patterns from data and achieving state-of-the-art but fail short to directly transfer across different graphs. Instead, it requires the cost of extensive training and hyperparameter optimization to adapt to the target graph. In this work, we introduce the Universal Link Predictor (UniLP), a novel model that combines the generalizability of heuristic approaches with the pattern learning capabilities of parametric models. UniLP is designed to autonomously identify connectivity patterns across diverse graphs, ready for immediate application to any unseen graph dataset without targeted training. We address the challenge of conflicting connectivity patterns—arising from the unique distributions of different graphs—through the implementation of In-context Learning (ICL). This approach allows UniLP to dynamically adjust to various target graphs based on contextual demonstrations, thereby avoiding negative transfer. Through rigorous experimentation, we demonstrate UniLP's effectiveness in adapting to new, unseen graphs at test time, showcasing its ability to perform comparably or even outperform parametric models that have been finetuned for specific datasets. Our findings highlight UniLP's potential to set a new standard in link prediction, combining the strengths of heuristic and parametric methods in a single, versatile framework.

## 1 Introduction

Graph-structured data is ubiquitous across diverse domains, including social networks (Liben-Nowell & Kleinberg, 2003), protein-protein interactions (Szklarczyk et al., 2019), movie recommendations (Koren et al., 2009), and citation networks (Yang et al., 2016). It encapsulates the complex relationships among entities, serving as a powerful data structure for analytical exploration. At the heart of graph analysis lies the task of link prediction (LP) (Yang et al., 2015; Dong et al., 2022; Guo et al., 2022), a crucial problem aimed at forecasting missing or future connections within these networks. Over the years, the quest to enhance LP accuracy has advanced the development of numerous methodologies (Kumar et al., 2020), broadly categorized into two main classes of approaches.

The first line of works is non-parametric heuristics link predictors, including Common Neighbor (CN) (Liben-Nowell & Kleinberg, 2003), Preferential Attachment (PA) (Barabási & Albert, 1999), Resource Allocation (RA) (Zhou et al., 2009) and Katz index (Katz, 1953). By discovering and abstracting the universal structural properties underlying different graphs (Barabási & Albert, 1999; Watts & Strogatz, 1998; Holland et al., 1983), heuristics methods are developed based on observing the connectivity patterns existing in real-world graph datasets. For example, CN assumes the tendency of triadic closure (Easley et al., 2010), such that a friend's friend is likely to be friends in a social network. These heuristics link predictors can be readily applied to any graph dataset with great generalizability. However, this approach relies on predefined heuristics, crafted from human expertise into the graph connectivity. Despite the initial success via capturing one specific connectivity pattern, they fail to capture all the effective structural features in the link prediction, leading to suboptimal performance when applied indiscriminately.

The other line of works is parametric link predictors, which automatically learn the connectivity patterns by fitting the LP models to the target graphs. These parametric methods, especially those Graph Neural Networks (Kipf & Welling, 2017; Hamilton et al., 2018) for Link Prediction (GNN4LP), have dominated the leaderboard of the link prediction tasks (Hu et al., 2021). Typically, these GNN4LP are provably the most expressive models such that the link representation is permutation-invariant (Zhang et al., 2021). They can capture more effective structural features compared to the simpler heuristics counterparts. However, their dependency on extensive training for each new graph dataset and the necessity for hyperparameter optimization (Chamberlain et al., 2022; Wang et al., 2023b; Dong et al., 2023) present notable challenges for their application across diverse graph environments.

Given that (1) the heuristics methods can be readily applied to any graphs without training based on common connectivity patterns and (2) the parametric model can automatically capture the connectivity patterns by fitting on the graph, a natural question arises:

> ***Can a singular LP model automatically learn and apply the connectivity pattern across new, unseen graphs without the need for direct training?***

An affirmative response would not only pioneer a new frontier in graph machine learning but also align with the transformative potential observed in foundation models across text and image processing fields (Brown et al., 2020; Kirillov et al., 2023). These models' exceptional generalizability, driven by their capability of transfer learning (Yosinski et al., 2014), offers a blueprint for the development of a universal LP model capable of broad applicability without explicit fitting.

**Present work.** In this study, we introduce the Universal Link Predictor (UniLP), a novel model designed for immediate application across diverse non-attributed graph environments[1] without the prerequisite of model fitting. Our investigation starts by assessing whether existing LP models possess the capability to transfer connectivity pattern knowledge from one graph to another. Through empirical and theoretical analyses spanning both heuristic and parametric link predictors, we uncover a significant challenge: negative transfer (Wang et al., 2021a) can happen when directly transferring the connectivity patterns across distinct graph datasets, including both real-world and synthetic examples. This complexity arises from the inherent diversity and flexibility of graph data, leading to unique connectivity patterns for each graph.

To equip UniLP with the capability to adapt to diverse graphs without the need for training, we are inspired by the concept of In-context Learning (ICL) as utilized by large language models (LLMs) (Brown et al., 2020). ICL enables models to adapt to new datasets or tasks through the relevant demonstration examples (Wang et al., 2023a). Analogously, for adapting our LP model to a particular graph, we select a collection of in-context links to act as such demonstration examples. These in-context links not only provide a context for link prediction but also aid in capturing the unique connectivity pattern inherent to the graph in question. To achieve link representations that are conditioned on the graph's specific connectivity pattern, we employ an attention mechanism (Vaswani et al., 2017; Brody et al., 2022). This mechanism facilitates dynamic

---

[1]In this study, we focus on non-attributed graphs. This choice is informed by previous findings indicating that node attributes have minimal impact on the effectiveness of LP tasks when compared to the structure of the graph. This is discussed in detail in Appendix E of (Dong et al., 2023).

adjustment of link representations in response to the graph context, enabling the model to accurately reflect the unique connectivity patterns of each graph.

We have curated a diverse collection of graph datasets spanning multiple domains, providing a rich variety of connectivity patterns for benchmarking. Through extensive experiments on these datasets, we demonstrate the seamless applicability of UniLP to novel and unseen graph datasets without requiring dataset-specific fitting. Notably, UniLP, empowered with ICL, exhibits the capability to meet or even exceed the performance levels of LP models that have been pretrained and finetuned for specific target graphs. This achievement underscores UniLP's broad applicability and robust adaptability, establishing a groundbreaking approach to link prediction tasks.

In summary, our contributions to the field of link prediction are:

- We pioneer in highlighting the challenges of applying a singular LP model across various graph datasets due to conflicting connectivity patterns, a finding supported by both empirical evidence and theoretical analysis.

- Addressing these challenges, we introduce UniLP, a novel LP approach leveraging ICL for dynamic adaptation to new graphs in real-time, thereby eliminating the need for traditional training processes.

- The diverse collection of graph datasets we've collected facilitates extensive validation of UniLP's adaptability. Our experiments confirm that UniLP is not only capable of adjusting to any new graph dataset during inference but also achieves competitive performance, marking a significant advancement in link prediction methodologies.

## 2  Can one model fit all?

Machine learning models perform a task by learning from data. The quest for generalizability in machine learning models has led to significant advancement in domains such as natural language processing (NLP) (Vaswani et al., 2017; Brown et al., 2020) and computer vision (CV) (Betker et al.; Kirillov et al., 2023). Foundation models in these fields have demonstrated remarkable generalizability across unseen datasets (Donahue et al., 2014), primarily due to their training on extensive data, which enables them to learn transferable knowledge.

In the context of LP tasks, the heuristics link predictors can be seen as a type of *transferable knowledge*. These predictors, crafted by manually analyzing common connectivity patterns in real-world graphs, offer insights into the underlying structure of networks. However, the validity of applying these heuristics universally is questioned, especially considering the wide spectrum of graph data. For instance, social networks like Facebook often exhibit a community-oriented structure (Newman, 2006a). Conversely, networks adhering to a scale-free power-law distribution (Barabási & Albert, 1999), such as the World Wide Web, tend to favor a Preferential Attachment connectivity pattern. Through both empirical and theoretical examination, we aim to explore the challenges posed by the direct application of connectivity patterns from one graph to another. Our findings will reveal that such an approach may lead to negative transfer (Wang et al., 2021a), emphasizing the critical need for adaptable strategies in the face of graph diversity.

### 2.1  Empirical evaluation on transferability

Our exploration begins with an empirical investigation aimed at understanding the transferability of learned connectivity patterns across diverse graph domains. We curate a collection of real-world graphs from varied fields in Table 1, ensuring comprehensive illustrations of different graph types.

To assess the potential of the important connectivity pattern, learned from one graph, to influence the LP performance on another, we incorporate extra graphs into the training phase of the target graph. In other words, this experiment deviates from the standard supervised learning approach by introducing additional training signals from other graphs. If the connectivity patterns from these extra graphs align with or augment the structure of the target graph, the LP model's performance should either remain stable or improve. To

make the experiment tractable, we only introduce **one additional** graph into the training graph and then make a link prediction on the target graph. This additional graph is kept disconnected from the target graph to ensure that the test set remains the same as standard LP tasks. In these experiments, we employ SEAL (Zhang & Chen, 2018) as the backbone model for the experiment and adopt Hits@50 as the performance metrics (Hu et al., 2021).

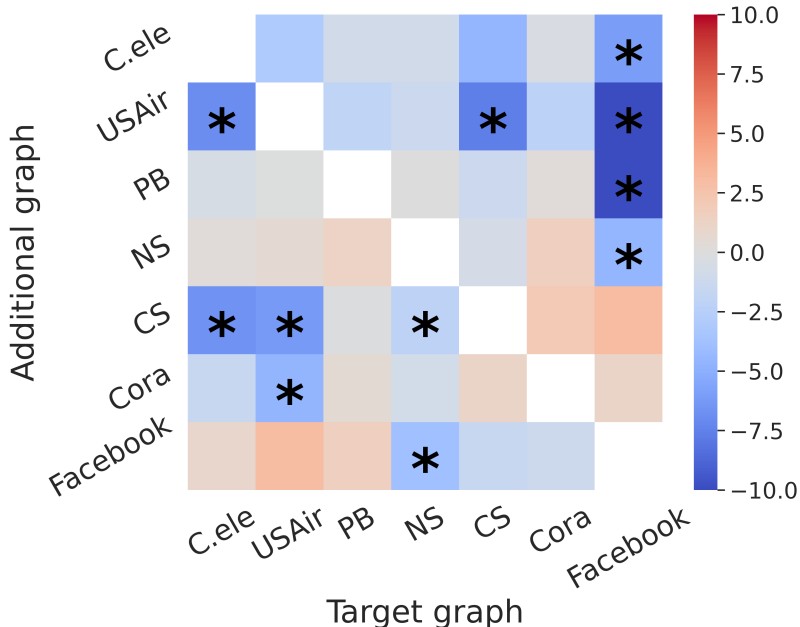

Figure 1: Performance change of SEAL (Zhang & Chen, 2018) after training with one additional graph. ✱denotes statistically significant change.

Results are presented in Figure 1. It shows how the LP model's performance is affected by the introduction of additional graph data during training. In the heatmap, warm colors represent an improvement in LP performance, while cooler colors denote a performance decrease. The predominance of cooler colors in the heatmap reveals that integrating an extra graph into training generally results in performance degradation. This observation underscores the potential discordance in the underlying characteristics of different graphs, leading to conflicts between the learned connectivity patterns . This phenomenon highlights the inherent challenge in deploying a singular LP model across various graphs, thus questioning the feasibility of a "one model fits all" approach in the context of LP tasks.

## 2.2 Conflicting patterns across graphs

In this section, we delve into the theoretical aspects of how the unique characteristics of different graphs can hinder the transferability of connectivity patterns. We begin with a formal definition and preliminary discussion of LP.

**Preliminary.** Consider an undirected graph $G = (V, E^o)$. $V$ is the set of nodes with size $n$, which can be indexed as $\{i\}_{i=1}^n$. $E^o$ denotes the *observed* set of links, which is a subset $E^o \subseteq E^*$ of the complete set of true links $E^* \subseteq V \times V$. Here, $E^*$ encompasses not only the observed links but also potential links that are currently absent or may form in the future within the graph $G$. For any node $v \in V$, $\mathcal{N}(v) = \{u | (u, v) \in E^o\}$ denotes the neighbors of node $v$. The set of $k$-hop simple paths from node $u$ to $v$ is denoted as $\pi_k(u, v) = \{(v_1, v_2, \ldots, v_k) | v_1 = u, v_k = v \text{ and } (v_i, v_{i+1}) \in E^o \text{ for } i \in \{1, \ldots, k-1\}\}$. Note that paths only contain distinct nodes. We denote the shortest-path between a node pair $(u, v)$ as $\text{SP}(u, v)$.

The objective of LP tasks is to identify the set of unobserved true links $E^u \subseteq E^* \backslash E^o$ within a given graph $G$. This task diverges from typical binary classification problems, as the potential candidates for $E^u$ are

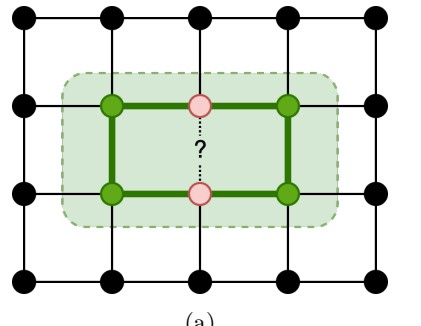 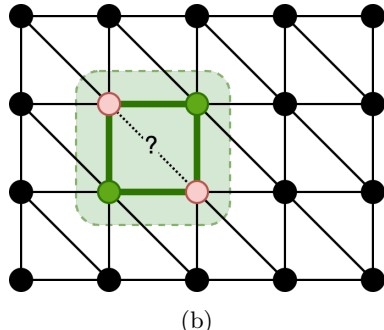

(a) (b)

Figure 2: Two synthetic graphs with different connectivity patterns: (a) **Grid** lattice graph; (b) **Triangular** lattice graph.

predetermined: they consist of all node pairs not already included in the observed links $V \times V \backslash E^o$. In practical terms, "identifying" $E^u$ equates to ranking these unobserved true links higher than false links based on their link features (Yang et al., 2015; Hu et al., 2021). This ranking process is defined by what we term *connectivity patterns*[2]:

**Definition 2.1. Connectivity pattern** is an ordered sequence of events $\omega = [A_1, A_2, \dots]$ such that $p(y = 1|A_i) \geq p(y = 1|A_j)$ for any $i < j$.

Here, an event $A$ refers to a specific set of conditions met by the link features of a node pair. In LP tasks, connectivity patterns may be determined by human experts using heuristic methods or by training parametric link predictors. For example, in social networks, a simple connectivity pattern might be $\omega = [CN(u, v) \geq 1, CN(u, v) = 0]$, suggesting that pairs of users with common friends are more likely to connect than those without any.

The ability to transfer a connectivity pattern from one graph to another suggests the potential for LP models to be applicable to new, previously unseen graphs. However, a mismatch in the ranking of connectivity patterns between the training and target graphs could lead to inaccuracies, since the model can assign higher scores to unlikely links and lower scores to likely ones.

Next, we demonstrate that even structurally similar synthetic graphs can exhibit different connectivity patterns. We begin by considering two types of lattice graphs: a **Grid** graph, similar to a chessboard, where nodes are evenly spaced on a 2D grid, each connected to its four nearest neighbors; and a **Triangular** graph, derived from the Grid by adding one diagonal edge within each square unit. Despite their structural similarities, these graphs, Grid and Triangular, display divergent connectivity patterns:

**Theorem 2.2.** *Define $A_2 = |\pi_2(u, v)| \geq 1$ and $A_3 = |\pi_3(u, v)| \geq 1$ as elements of $\omega$. The connectivity patterns on Grid and Triangular graphs are distinct. Specifically:*
*(i) On Grid: $\omega = [A_3, A_2]$; (ii) On Triangular: $\omega = [A_2, A_3]$.*

The proof is in Appendix C.2. In essence, in Triangular graphs, node pairs two hops away are more likely to form a link compared to those three hops away. Conversely, in Grid graphs, despite their structural similarity to Triangular graphs, node pairs two hops away have no likelihood of linking.

This observation of conflicting connectivity patterns across similar graphs underlines the challenges in knowledge transfer for LP tasks. Even slight structural variations in graphs can significantly alter the likelihood of link formation between nodes. Consequently, the task of developing a universal link predictor, capable of adapting to any graph without specific tuning for its connectivity pattern, is a non-trivial endeavor.

---

[2]We have an in-depth analysis on how connectivity patterns differ from and relate to graph distributions in Appendix C.1, where we illustrate that graphs with different underlying distribution could have the shared connectivity pattern.

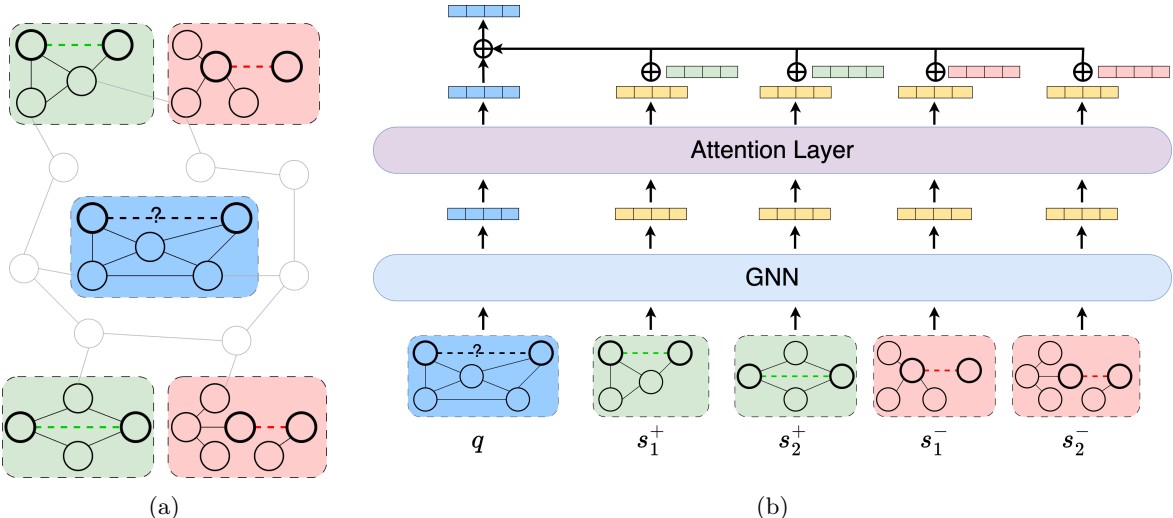

Figure 3: Overview of the UniLP framework. (a) For predicting a query link $q$, we initially sample positive $(s^+)$ and negative $(s^-)$ in-context links from the target graph. (b) Both the query link and these in-context links are independently processed through a shared subgraph GNN encoder. An attention mechanism then calculates scores based on the similarity between the query link and the in-context links. The final representation of the query link, contextualized by the target graph, is obtained through a weighted summation, which combines the representations of the in-context links with their respective labels.

## 2.3 Contextualizing Link Prediction

The challenge of conflicting connectivity patterns across different graphs highlights a critical issue: a model trained on one graph may break down when applied to another without accommodating the unique characteristics of the target graph. To mitigate this, we suggest a paradigm where the model dynamically adapts to the target graph by taking into account its specific characteristics.

This adjustment process involves conditioning the model on the target graph's properties, thereby ensuring that the prediction of link formation, $p(1|A)$, is influenced not just by the inherent link features but also by the properties of the target graph. We draw inspiration from the concept of In-context Learning (ICL) in LLMs (Dai et al., 2022), which enables LLMs to solve tasks with a few demonstration examples. We propose the incorporation of the target graph as a contextual element $c$ in the link prediction $p(1|A, c)$. By doing so, the model learns to understand the joint distribution of link features and the graph context, allowing it to adapt to different graphs. In the subsequent section, we will delve into the practical implementation of an LP model equipped with ICL capabilities.

## 3 Universal Link Predictor

This section outlines our proposed UniLP, designed for effective application to unseen datasets. UniLP can readily adapt to the connectivity patterns of any new graphs, even though they are not seen during the training phase. UniLP operates by first sampling a set of in-context links from the target graph, which are then independently encoded alongside the target link using a shared GNN encoder. An attention mechanism is employed to merge the representations of these in-context links in relation to their interaction with the target link, forming a composite representation for the final prediction. The overall framework is in Figure 3.

### 3.1 Query and in-context links

For a given target link $q \in V \times V$ in graph $G$, we define it as the *query* link. To predict this link based on the contextual information of $G$, we start by sampling a set of *in-context* links from $G$. Specifically, we select $k$ node pairs $S^+ \subseteq E^o$ as positive examples, where $S^+ = \{s_1^+, s_2^+, \ldots, s_k^+\}$. These pairs have existing

links between them in $G$. Similarly, we gather negative examples $S^- = \{s_1^-, s_2^-, \ldots, s_k^-\} \subseteq V \times V \backslash E^o$, comprising $k$ node pairs without a link. The combined set $S^+ \cup S^-$ approximates the overall properties of $G$ and provides a context $c$ for the model to perform LP $(p(1|A, c))$ using both link features and graph context.

Once we get the query link and the in-context links, we need to obtain the structural representation for them. We start by extracting the ego-subgraph for each of them. An ego-subgraph $\mathbb{G}((u, v), r, G)$ for a node pair $(u, v)$ is a subgraph induced by all the $r$-hop neighboring nodes of the nodes $u$ and $v$ on the graph $G$:

$$\mathbb{G}((u, v), r, G) = (V_s, E_s),$$

where $V_s = \{i | \text{SP}(i, u) \leq r \text{ or } \text{SP}(i, v) \leq r\}$ and $E_s = \{(i, j) \in E^o | i, j \in V_s\}$. For simplicity, we denote such an ego-subgraph as $\mathbb{G}(e)$ for the node pair $e = (u, v)$ when there is no ambiguity. The ego-subgraphs for the query link and the in-context links are $\{\mathbb{G}(e) | e \in \{q\} \bigcup S^+ \bigcup S^-\}$.

Utilizing ego-subgraphs to represent links offers several key advantages over using either individual node pairs or the entire graph. Firstly, an ego-subgraph provides a richer structural context than a mere node pair, encapsulating the local neighborhood structure around the link in question. This approach allows for a more detailed and informative representation of the link's local structures. Secondly, ego-subgraphs serve as an effective and computationally efficient approximation of global link features (Zhang & Chen, 2018). This is advantageous over the resource-intensive process of encoding the entire graph. Lastly, representations at the subgraph level are inherently more expressive compared to node-level representations (Frasca et al., 2022). This enhanced expressiveness is crucial for capturing the structures of links and performing accurate LPs.

## 3.2 Encoding ego-subgraphs

The ego-subgraphs within the set $\{\mathbb{G}(e) | e \in \{q\} \cup S^+ \cup S^-\}$ vary in size, requiring a uniform approach to representation. We employ GNNs to encode these subgraphs into a consistent latent space.

In the absence of node features in non-attributed graphs, typical GNNs (Kipf & Welling, 2017; Hamilton et al., 2018; Xu et al., 2018) require initial input vectors for each node. Common methods like assigning identical or random vectors meet this requirement but lack expressiveness about graph structures (Li et al., 2020; Zhang et al., 2021). To address this, we utilize the *labeling trick* technique, assigning each node $i$ in $\mathbb{G}(e)$ a positional encoding based on its relative position to the target link $e = (u, v)$. A proper design of labeling trick can ensure that GNNs are expressive enough to capture the heuristic graph substructures, like Common Neighbor and Shortest Path.

We propose *DRNL+*, which extends the labeling tricks applied in Double Radius Node Labeling (DRNL) (Zhang & Chen, 2018) and Distance Encoding (DE) (Li et al., 2020). In DRNL, nodes $i$ in $\mathbb{G}(e)$ are assigned integer labels as follows:

$$\text{DRNL}(i, (u, v)) = 1 + \min(d_u, d_v) + (d/2)[(d/2) + (d\%2) - 1],$$

where $d_u := \text{SP}(i, u)$, $d_v := \text{SP}(i, v)$, and $d := d_u + d_v$. However, DRNL doesn't distinguish nodes reachable to only one of the target nodes. Thus, *DRNL+* enhances this by using DE to assign a tuple of integers:

$$\text{DRNL+}(i, (u, v)) = \begin{cases} (0, d_u), & \text{if } d_v = \infty \\ (0, d_v), & \text{if } d_u = \infty \\ (\text{DRNL}(i, (u, v)), 0), & \text{otherwise} \end{cases} \tag{1}$$

After the labeling trick indicates relative positions, we apply the SAGE (Hamilton et al., 2018) with mean aggregation to update node representations (Zeng et al., 2022). The final subgraph representation, $h_e \in \mathbb{R}^F$ for each link $e \in \{q\} \bigcup S^+ \bigcup S^-$, is derived by average pooling the representations of all nodes in $\mathbb{G}(e)$. We find that mean aggregation and pooling work best for such a universal link predictor, hypothesizing that this approach better accommodates varying graph sizes and node degrees, thereby enhancing model generalizability.

### 3.3 Link prediction with context

Once the ego-subgraphs are encoded into latent space, we utilize these representations $\{h_e | e \in \{q\} \cup S^+ \cup S^-\}$ to parameterize our link predictor $p(1|A, c)$ via an attention mechanism (Vaswani et al., 2017).

The attention scores $a$ between the query link representation $h_q$ and each in-context link representation $h_s$ for $s \in S^+ \cup S^-$ are calculated using additive attention (Niu et al., 2021; Brody et al., 2022):

$$a_s = p^\top \text{LeakyReLU}\left(\boldsymbol{W}_k \cdot [h_q \| h_s]\right), \tag{2}$$

where $p^\top \in \mathbb{R}^{F'}$ is a learnable vector and $\boldsymbol{W}_k \in \mathbb{R}^{F' \times 2F}$ is a projection matrix. The concatenation operation is denoted by $\|$. The normalized attention scores $\alpha$ are obtained as follows:

$$\alpha_s = \text{softmax}\left(a_s\right) = \frac{\exp\left(a_s\right)}{\sum_{e \in S^+ \bigcup S^-} \exp\left(a_e\right)}. \tag{3}$$

We denote the attention score between the query link and a positive in-context link $s^+ \in S^+$ as $\alpha^+$, and with a negative in-context link as $\alpha^-$. Like in Transformer and GAT models (Veličković et al., 2018; Brody et al., 2022), multi-head attention can also be employed to capture diverse interactions between graph structures.

**Remark.** The attention scores are pivotal for shaping the query link's representation in the context of the target graph. We intentionally exclude label information from the attention score computation to avoid biasing the model towards easy predictions during training. This approach aligns with an "unsupervised" learning strategy, as opposed to a "supervised" one, where label information might lead the model to rely excessively on seen patterns, thus turning the attention mechanism into a *de facto* classifier. This could hinder the model's ability to generalize and adapt across varying graph structures, increasing the risk of overfitting to specific connectivity patterns not applicable to new, unseen graphs. Our empirical findings support this methodology, demonstrating that keeping the attention computation label-free significantly boosts the model's generalizability.

After the normalized attention scores $\alpha$ are determined, we compute the final representation for the query link $q$. This is achieved by applying a weighted sum to the representations of the in-context links, using the attention scores as weights. Additionally, we integrate label information into the in-context links' representations by adding corresponding learnable vectors. Formally, the final representation is calculated as follows:

$$\tilde{h}_q = \sum_{s \in S^+} \alpha_s^+ \boldsymbol{W}_v \left(h_s + l^+\right) + \sum_{s \in S^-} \alpha_s^- \boldsymbol{W}_v \left(h_s + l^-\right), \tag{4}$$

where $l^+, l^- \in \mathbb{R}^{F'}$ are learnable vectors for labels, and $\boldsymbol{W}_v \in \mathbb{R}^{F' \times F}$ is a value projection matrix. The representation $\tilde{h}_q$ encapsulates both the link features of the query link $q$ and an estimation of the target graph $G$, and is then input into an MLP classifier to produce the link prediction result:

$$p(1|A, c) = \sigma\left(\text{MLP}\left(\tilde{h}_q\right)\right), \tag{5}$$

where $\sigma\left(\cdot\right)$ denotes a sigmoid function.

### 3.4 Pretraining objective

The pretraining objective for UniLP focuses on predicting the query link $q$ based on its own features and the context of the graph it is part of. We align this objective with standard binary classification as seen in typical parametric link prediction algorithms (Zhang & Chen, 2018; Chamberlain et al., 2022; Dong et al., 2023). In this setting, the classification label $y_e$ for an edge $e$ is set to 1 if $e$ is among the observed links $E^o$; otherwise, $y_e$ is 0. Additionally, we consider a set of pretrain graphs $\mathcal{G}$, with each graph $G$ being a member of this set. The overall pretraining loss is then defined as:

$$\mathcal{L} = \mathbb{E}_{G \in \mathcal{G}, e \in V \times V} \text{BCE}\left(\text{MLP}\left(\tilde{h}_e\right), y_e\right). \tag{6}$$

This loss function is employed across multiple graphs, allowing UniLP to learn a generalizable pattern for link prediction across various graph structures.

## 4 Related work

**Link prediction.** Traditional LP methods are handcrafted heuristics designed by observing the connectivity pattern in real-world data. They leverage either the link's local (Liben-Nowell & Kleinberg, 2003; Adamic & Adar, 2003; Zhou et al., 2009; Barabási & Albert, 1999) or global information (Katz, 1953; Page et al., 1999) to infer the missing links in the graph. OLP (Ghasemian et al., 2020) stacks the heuristics link predictors as a feature vector and fits a random forest as the classifier. WLNM (Zhang & Chen, 2017) is one of the pioneers in training a neural network as a link predictor. GAE (Kipf & Welling, 2016), as the first GNN4LP, utilizes GNNs to encode the graph structure into node representation and perform the link prediction task. SEAL (Zhang & Chen, 2018; Zhang et al., 2021) points out that a link-level representation is necessary for a successful LP method and proposes the labeling trick to enable GNNs to learn the joint structural representation. GraIL (Teru et al., 2020) extends the idea of the node labeling trick (Zhang et al., 2021) of SEAL to the knowledge graph completion tasks. NBFNet (Zhu et al., 2021) proposes a general framework for performing link prediction based on learnable paths. ELPH (Chamberlain et al., 2022), NCNC (Wang et al., 2023b), and MPLP (Dong et al., 2023) further improve the scalability of GNN4LP and achieve the state-of-the-arts on various graph benchmarks. On heterogeneous graphs, SLiCE (Wang et al., 2021b) adopts a pretrain-and-finetune strategy to learn contextual node representations for link predictions. Daza et al. (2021) studies how to perform inductive link prediction over knowledge graphs based on the textual data of nodes.

**In-context Learning.** The remarkable efficacy of LLMs across a broad spectrum of language tasks is significantly attributed to their adeptness in ICL (Brown et al., 2020). This capability allows LLMs to generalize to new tasks by leveraging demonstration examples, effectively learning the required skills on the fly. Irie et al. (2022) delves into the equivalence between conventional model training and the application of attention mechanisms to training samples during inference, suggesting an underlying mechanism of ICL. Further exploration by Dai et al. (2022) posits that ICL facilitates an implicit optimization process guided by in-context examples. While the concept of ICL has been primarily associated with LLMs, Prodigy (Huang et al., 2023) represents an initial attempt to adapt ICL for GNN-based models. Their approach, however, is somewhat constrained by the overlap in pretrain and test datasets, which raises questions about the method's transferability across distinct graph domains.

## 5 Experiments

In this section, we conduct extensive experiments to assess the performance of UniLP on new unseen datasets.

### 5.1 Experimental setup

**Benchmark datasets.** The foundation for our model's training is a collection of graph datasets spanning a variety of domains. Following (Mao et al., 2023), we have carefully selected graph data from fields such as biology (Von Mering et al., 2002; Zhang et al., 2018; Watts & Strogatz, 1998), transport (Watts & Strogatz, 1998; Batagelj & Mrvar, 2006), web (Ackland & others, 2005; Spring et al., 2002; Adamic & Glance, 2005), academia collaboration (Shchur et al., 2019; Newman, 2006b), citation (Yang et al., 2016), and social networks (Rozemberczki et al., 2021). This diverse selection ensures that we can pretrain and evaluate the LP model based on a wide array of connectivity patterns. The details of the curated graph datasets can be found in Table 1.

**Baseline Methods.** We compare UniLP with both heuristic and GNN-based parametric link predictors. Heuristic methods include Common Neighbor (CN) (Liben-Nowell & Kleinberg, 2003), Adamic-Adar index (AA) (Adamic & Adar, 2003), Resource Allocation (RA) (Zhou et al., 2009), Preferential Attachment (PA) (Barabási & Albert, 1999), Shortest-Path (SP), and Katz index (Katz) (Katz, 1953). GNN-based methods include GAE (Kipf & Welling, 2016), SEAL (Zhang & Chen, 2018), ELPH (Chamberlain et al., 2022), NCNC (Wang et al., 2023b), and MPLP (Dong et al., 2023). For GAE and NCNC, which require initial node features, we use a 32-dimensional all-one vector. All other methods can handle non-attributed graphs directly.

Table 1: The pretrain datasets and test benchmarks.

| Dataset | Pretrain | Test | # Nodes | # Edges | Avg. node deg. | Std. node deg. | Max. node deg. | Density |
|---------|----------|------|---------|---------|----------------|----------------|----------------|---------|
| | | | | | Biology | | | |
| Ecoli | ✓ | - | 1805 | 29320 | 16.24 | 48.38 | 1030 | 1.8009% |
| Yeast | ✓ | - | 2375 | 23386 | 9.85 | 15.5 | 118 | 0.8295% |
| Celegans | - | ✓ | 297 | 4296 | 14.46 | 12.97 | 134 | 9.7734% |
| | | | | | Transport | | | |
| Power | ✓ | - | 4941 | 13188 | 2.67 | 1.79 | 19 | 0.1081% |
| USAir | - | ✓ | 332 | 4252 | 12.81 | 20.13 | 139 | 7.7385% |
| | | | | | Web | | | |
| PolBlogs | ✓ | - | 1490 | 19025 | 12.77 | 20.73 | 256 | 1.7150% |
| Router | ✓ | - | 5022 | 12516 | 2.49 | 5.29 | 106 | 0.0993% |
| PB | - | ✓ | 1222 | 33428 | 27.36 | 38.42 | 351 | 4.4808% |
| | | | | | Collaboration | | | |
| Physics | ✓ | - | 34493 | 495924 | 14.38 | 15.57 | 382 | 0.0834% |
| CS | - | ✓ | 18333 | 163788 | 8.93 | 9.11 | 136 | 0.0975% |
| NS | - | ✓ | 1589 | 5484 | 3.45 | 3.47 | 34 | 0.4347% |
| | | | | | Citation | | | |
| Pubmed | ✓ | - | 19717 | 88648 | 4.5 | 7.43 | 171 | 0.0456% |
| Citeseer | ✓ | - | 3327 | 9104 | 2.74 | 3.38 | 99 | 0.1645% |
| Cora | - | ✓ | 2708 | 10556 | 3.9 | 5.23 | 168 | 0.2880% |
| | | | | | Social | | | |
| Twitch | ✓ | - | 34118 | 429113 | 12.58 | 35.88 | 1489 | 0.0737% |
| Github | ✓ | - | 37700 | 289003 | 7.67 | 46.59 | 6809 | 0.0407% |
| Facebook | - | ✓ | 22470 | 171002 | 7.61 | 15.26 | 472 | 0.0677% |

**Evaluation of UniLP**  To evaluate UniLP's effectiveness on unseen datasets, we divide our graph data into non-overlapping pretrain and testing sets (see Table 1) and pretrain one single model on the combined pretrain datasets. During pretraining, we dynamically sample 40 positive and negative links as in-context links $S^+ \cup S^-$ for each query link from the corresponding pretrain dataset. For evaluation, each test dataset is split into 70%/10%/20% for training/validation/testing. The training set here forms the observed links $E^o$, while validation and test sets represent unobserved links $E^u$. During the inference, we sample $k = 200$ positive and negative links as in-context links per test dataset. We report Hits@50 (Hu et al., 2021) as the evaluation metric for LP. More details about the pretraining of UniLP can be found in Appendix A.1.

**Evaluation of Baselines.**  Baseline models follow similar evaluation procedures, with adaptations for transfer learning capabilities. We employ two settings: (1) **Pretrain Only**, where models are trained on combined pretrain datasets  and then tested on each test dataset, and (2) **Pretrain & Finetune**, where after pretraining, models are additionally finetuned on each test dataset with 200 sampled positive and negative links for training.

## 5.2  Primary results

Table 2 presents the performance of UniLP on various unseen graph datasets. The results demonstrate that UniLP outperforms both traditional heuristic methods and standard GNN-based LP models that are pre-trained without specific adaptation, showing significant improvements in 4 out of 7 the benchmark datasets. This performance enhancement suggests that tailoring the LP model to individual graphs can markedly increase its transfer learning capabilities.

Moreover, UniLP achieves comparable or even superior results to GNN-based LP models that undergo finetuning,  despite not being explicitly trained on the test data. This highlights the effectiveness of the ICL capability in UniLP, which allows the model to adapt seamlessly to specific graph datasets without the need for additional training. By leveraging in-context links provided during the inference phase, UniLP

Table 2: Link prediction results on test datasets evaluated by Hits@50. The format is average score ± standard deviation. The top three models are colored by **First**, **Second**, **Third**.

| | Biology | Transport | Web | Collaboration | | Citation | Social | |
|---|---|---|---|---|---|---|---|---|
| | **C.ele** | **USAir** | **PB** | **NS** | **CS** | **Cora** | **Facebook** | **Ave. Rank** |
| Heuristics | | | | | | | | |
| **CN** | $46.88_{\pm 12.28}$ | $82.75_{\pm 1.54}$ | $41.15_{\pm 3.77}$ | $74.03_{\pm 1.59}$ | $56.84_{\pm 15.56}$ | $33.85_{\pm 0.93}$ | $58.70_{\pm 0.35}$ | 11.00 |
| **AA** | $61.07_{\pm 5.16}$ | $86.96_{\pm 2.24}$ | $44.12_{\pm 3.36}$ | $74.03_{\pm 1.59}$ | $68.22_{\pm 1.08}$ | $33.85_{\pm 0.93}$ | $67.80_{\pm 2.12}$ | 5.71 |
| **RA** | $62.80_{\pm 4.84}$ | $87.27_{\pm 1.89}$ | $43.72_{\pm 2.86}$ | $74.03_{\pm 1.59}$ | $68.21_{\pm 1.08}$ | $33.85_{\pm 0.93}$ | $68.84_{\pm 2.03}$ | 5.57 |
| **PA** | $43.85_{\pm 4.12}$ | $77.69_{\pm 2.29}$ | $28.93_{\pm 1.91}$ | $35.35_{\pm 3.01}$ | $6.49_{\pm 0.61}$ | $22.09_{\pm 1.52}$ | $12.95_{\pm 0.63}$ | 13.71 |
| **SP** | $0.00_{\pm 0.00}$ | $0.00_{\pm 0.00}$ | $0.00_{\pm 0.00}$ | $80.00_{\pm 1.11}$ | $41.34_{\pm 35.58}$ | $52.97_{\pm 1.53}$ | $0.00_{\pm 0.00}$ | 13.14 |
| **Katz** | $58.86_{\pm 6.48}$ | $84.64_{\pm 1.86}$ | $44.36_{\pm 3.65}$ | $78.96_{\pm 1.35}$ | $66.32_{\pm 5.59}$ | $52.97_{\pm 1.53}$ | $60.79_{\pm 0.60}$ | 6.86 |
| Pretrain Only | | | | | | | | |
| **SEAL** | $61.28_{\pm 3.76}$ | $86.00_{\pm 1.56}$ | $45.44_{\pm 2.68}$ | $84.07_{\pm 1.96}$ | $62.82_{\pm 1.62}$ | $56.21_{\pm 2.24}$ | $54.57_{\pm 1.48}$ | 5.57 |
| **GAE** | $44.71_{\pm 3.39}$ | $76.12_{\pm 2.27}$ | $27.56_{\pm 2.34}$ | $15.20_{\pm 2.14}$ | $5.08_{\pm 0.48}$ | $24.22_{\pm 1.53}$ | $6.65_{\pm 0.57}$ | 14.14 |
| **ELPH** | $59.23_{\pm 4.50}$ | $84.42_{\pm 2.22}$ | $43.69_{\pm 2.90}$ | $84.27_{\pm 1.43}$ | $70.69_{\pm 3.63}$ | $56.91_{\pm 1.43}$ | $61.80_{\pm 2.46}$ | 5.57 |
| **NCNC** | $48.07_{\pm 4.79}$ | $75.44_{\pm 4.22}$ | $25.66_{\pm 1.42}$ | $80.07_{\pm 1.43}$ | $34.27_{\pm 2.28}$ | $52.51_{\pm 2.54}$ | $19.28_{\pm 1.58}$ | 11.57 |
| **MPLP** | $56.74_{\pm 5.31}$ | $82.94_{\pm 2.30}$ | $47.78_{\pm 2.55}$ | $80.33_{\pm 1.54}$ | $24.26_{\pm 1.28}$ | $46.71_{\pm 2.25}$ | $48.06_{\pm 2.09}$ | 8.86 |
| Pretrain & Finetune | | | | | | | | |
| **SEAL** | $64.45_{\pm 4.14}$ | $88.49_{\pm 2.16}$ | $47.78_{\pm 3.32}$ | $84.84_{\pm 2.32}$ | $61.54_{\pm 3.09}$ | $62.19_{\pm 3.27}$ | $58.70_{\pm 2.78}$ | 3.14 |
| **GAE** | $44.71_{\pm 4.07}$ | $74.47_{\pm 2.96}$ | $25.92_{\pm 2.64}$ | $18.34_{\pm 2.27}$ | $4.95_{\pm 0.44}$ | $25.31_{\pm 1.48}$ | $6.11_{\pm 0.39}$ | 14.71 |
| **ELPH** | $60.51_{\pm 5.72}$ | $84.52_{\pm 2.08}$ | $43.58_{\pm 3.48}$ | $86.08_{\pm 0.69}$ | $71.10_{\pm 3.48}$ | $57.18_{\pm 1.89}$ | $63.31_{\pm 3.63}$ | 4.71 |
| **NCNC** | $64.45_{\pm 5.10}$ | $85.85_{\pm 2.46}$ | $47.75_{\pm 6.90}$ | $88.25_{\pm 2.25}$ | $58.75_{\pm 5.74}$ | $60.00_{\pm 2.50}$ | $59.32_{\pm 6.93}$ | 3.71 |
| **MPLP** | $62.56_{\pm 4.79}$ | $85.08_{\pm 1.54}$ | $48.01_{\pm 2.94}$ | $80.16_{\pm 1.18}$ | $50.35_{\pm 1.01}$ | $56.02_{\pm 2.09}$ | $56.72_{\pm 1.20}$ | 6.14 |
| Ours | | | | | | | | |
| **UniLP** | $65.20_{\pm 4.40}$ | $85.98_{\pm 2.00}$ | $48.14_{\pm 2.99}$ | $89.09_{\pm 2.05}$ | $64.59_{\pm 2.65}$ | $57.50_{\pm 2.40}$ | $65.49_{\pm 2.05}$ | 1.86 |

can dynamically adjust its knowledge of connectivity patterns, demonstrating its potential to deliver robust performance across a wide range of unseen graph datasets. In addition, the experimental results on the synthetic Triangular/Grid lattice graphs can be found in Table 4 in the Appendix.

### 5.3 The inner mechanism of UniLP

Table 3: Link prediction results on test datasets evaluated by Hits@50 under context perturbation. This table presents the outcomes of link prediction when the context, i.e., in-context links, is deliberately altered. The aim is to analyze how changes in the context influence the final prediction accuracy.

| | Biology | Transport | Web | Collaboration | | Citation | Social |
|---|---|---|---|---|---|---|---|
| | **C.ele** | **USAir** | **PB** | **NS** | **CS** | **Cora** | **Facebook** |
| **UniLP-FlipLabel** | $0.61_{\pm 0.27}$ | $15.81_{\pm 13.17}$ | $0.03_{\pm 0.03}$ | $27.97_{\pm 4.29}$ | $0.60_{\pm 0.23}$ | $2.03_{\pm 0.55}$ | $0.32_{\pm 0.15}$ |
| **UniLP-RandomContext** | $52.89_{\pm 5.90}$ | $81.91_{\pm 2.14}$ | $47.47_{\pm 3.05}$ | $85.60_{\pm 1.23}$ | $47.80_{\pm 6.48}$ | $37.62_{\pm 5.64}$ | $22.17_{\pm 6.55}$ |
| **UniLP** | $65.20_{\pm 4.40}$ | $85.98_{\pm 2.00}$ | $48.14_{\pm 2.99}$ | $89.09_{\pm 2.05}$ | $64.59_{\pm 2.65}$ | $57.50_{\pm 2.40}$ | $65.49_{\pm 2.05}$ |

We further explore the capability of our proposed model's ICL to facilitate skill learning (Pan et al., 2023; Mao et al., 2024), enabling the model to acquire new skills not encountered during the pretraining phase, guided by ICL demonstrations. This investigation focuses on the model's performance sensitivity to corrupting in-context links, particularly when these links are presented with incorrect input-label associations. Given that each in-context link consists of an input and its corresponding label, we introduce two perturbation strategies to assess this sensitivity: **FlipLabel**: we invert the labels of the in-context links, labeling previously positive links as negative and vice versa. **RandomContext**: Instead of selecting in-context links from the target graph, we randomly sample them from a graph generated using the Stochastic Block Model (Holland et al., 1983).

The outcomes, as shown in Table 3, reveal that flipping the labels of in-context links significantly degrades the model's performance, rendering it almost ineffective. This finding underscores the model's utilization of ICL for skill learning, specifically in learning new feature-label mappings within a given context (Wei et al., 2023; Min et al., 2022). It highlights the pivotal role of accurate label information in in-context links for the model's effective adaptation to the target graph.

Furthermore, using randomly generated graphs as a source of in-context links also detrimentally affects performance, albeit to varying extents across different datasets. This implies the importance of choosing in-context links that genuinely represent the properties of the target graph. Interestingly, the less severe performance decline in some datasets may indicate their inherent community-based graph structures.

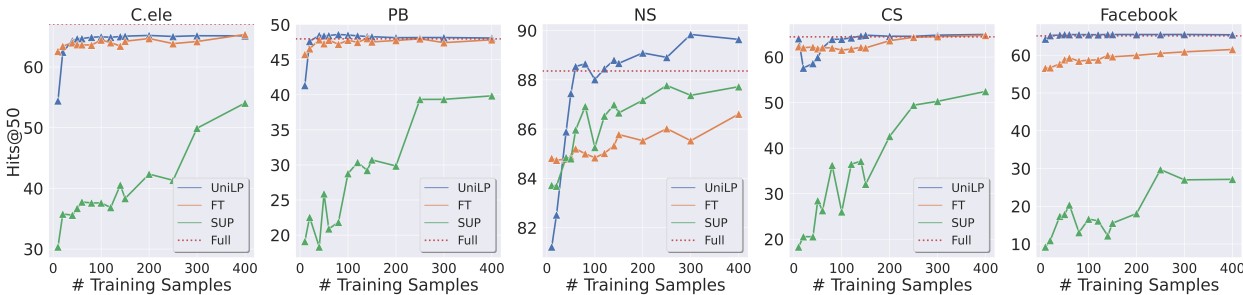

Figure 4: Performance of UniLP with varying quantities of in-context links.

## 5.4 Effectiveness of in-context links' size

This experiment evaluates how varying the quantity of in-context links affects UniLP's performance during inference. We experiment with different numbers of in-context links, ranging from 10 to 400, sampled from each test graph. These links are used as context for the model. Additionally, we utilize SEAL as the base model and assess its performance under finetuning (FT) and supervised training (SUP) from scratch with varying training sample sizes. For comparison, we also include results from training a SEAL model on the full set of target graph data (Full), as detailed in Figure 4.

The findings reveal a consistent improvement in UniLP's performance with an increasing number of in-context links. This indicates that our method can more effectively capture the target graph's properties with additional context. Notably, on four of the test datasets, UniLP either matches or exceeds the performance of models trained end-to-end on the entire graph. This suggests that leveraging more pretraining data can be advantageous for LP tasks when properly managed. Furthermore, despite both UniLP and the finetuned models being pretrained on the same datasets and using the same in-context links, UniLP can outperform its finetuned counterparts. This observation suggests that in some cases, utilizing ICL can be a more effective approach for adapting a pretrained model to a specific target dataset compared to finetuning. The trend on the rest of graphs can be found in Figure 7 in Appendix.

## 5.5 Visualization of the link representation

We conduct a comparative visualization of link representations as learned by a Pretrained Only SEAL model and UniLP. This comparison is shown in Figure 5. The results indicate that a naively pretrained model tends to map link representations from various graph datasets into a close subspace, potentially leading to indistinguishable link representations across different graphs, even when these graphs exhibit conflicting connectivity patterns.

In contrast, the link representations generated by UniLP, which are conditioned on the context of the target graph, demonstrate a distinct separation between different datasets. This separation is indicative of UniLP's effective ICL capability, which adeptly captures the subtle distributional differences across graphs. By

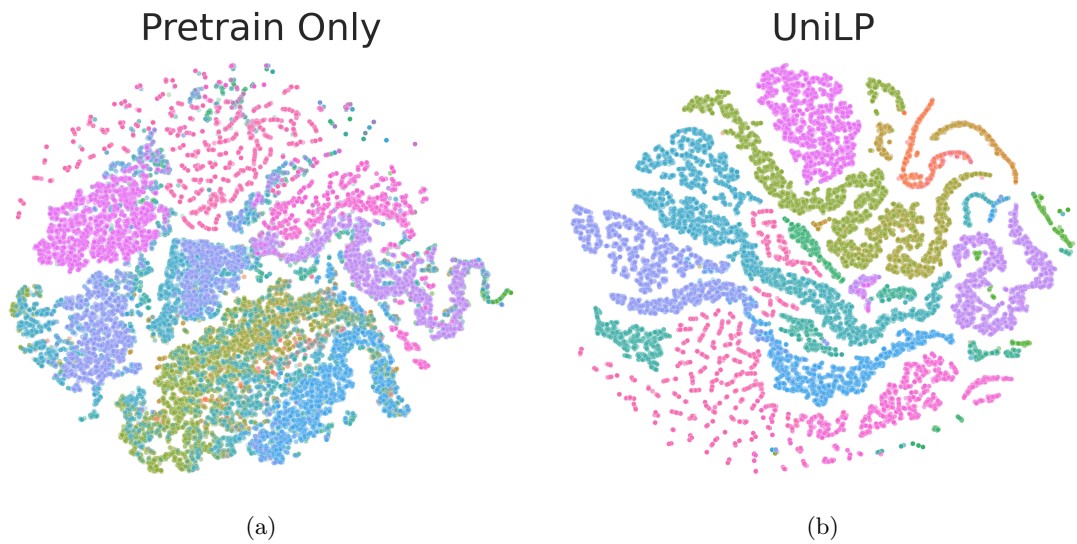

Figure 5: Visualization of the link representation learned from (a) Pretrain Only SEAL; (b) UniLP. Different colors indicate different test datasets.

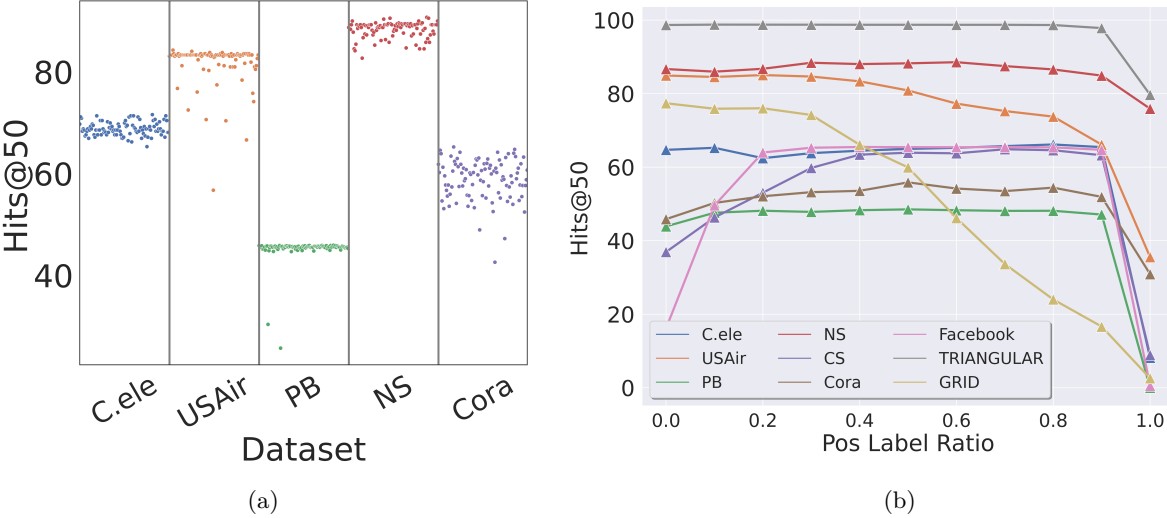

Figure 6: (a) Impact of Diverse In-Context Link Sets on LP Performance. This figure evaluates the variability in UniLP's performance across different sets of in-context links sampled. (b) Influence of positive-to-negative in-context link ratios on LP performance. The x-axis shows the ratio of positive to negative in-context links. A value of 0 indicates that all in-context links are negative, while a value of 1 indicates that all in-context links are positive.

adjusting the link representations based on the context provided by in-context links, UniLP can effectively address the challenge of conflicting connectivity patterns in diverse graph datasets.

## 5.6 Diversifying context

In our prior analysis, we utilized a fixed set of in-context links sampled from each target graph to serve as the context. This section delves into the impact of varying these in-context links by employing different

random seeds, aiming to discern the sensitivity of UniLP's performance to the specific selection of in-context links for each target graph. The outcomes of this investigation are shown in Figure 6a.

For the C.ele, USAir and PB datasets, the selection of in-context links has a minimal effect on UniLP's performance. However, for the NS and Cora datasets, the choice of in-context links significantly influences performance. The findings reveal that the selection of in-context links indeed affects UniLP's performance across the test datasets to varying extents, highlighting the importance of the selection process for these contextual links in optimizing the model's efficacy. This echoes the findings in the NLP domain, where large language models are sensitive to the selection of in-context demonstrations (Zhao et al., 2021). The study on the selection of the in-context links can be an interesting future work.

### 5.7 Varying positive-to-negative ratios of in-context links

In our previous exploration of LP performance, we initially maintain a balanced set of positive and negative in-context links for each query link during both pretraining and testing phases. This study delves into the effects of changing the ratio of positive to negative in-context links, while keeping the total count constant at 200, to assess the impact on LP accuracy. The findings, shown in Figure 6b, reveal several noteworthy observations.

Remarkably, for the majority of graph datasets examined, increasing the proportion of negative samples—contrary to intuitive expectations—does not detract from performance and, in some cases, matches the efficacy of a balanced distribution of in-context links. This phenomenon indicates that negative samples are more informative as positive ones for leveraging the ICL capabilities of UniLP. Specifically, in the case of the synthetic Grid graph, a higher ratio of negative samples significantly enhances LP performance, given a fixed total number of in-context links. This improvement may stem from the symmetry of positive link structures within the Grid graph, which exhibit a consistent connectivity pattern. The introduction of a greater variety of negative samples seems to enrich the model's learning context, effectively harnessing UniLP's ICL potential to capture more diverse disconnectivity patterns.

An exception to this trend is observed with the Facebook graph dataset, where a balance between positive and negative in-context links yields the most favorable outcomes. This suggests that for certain graph types, a balanced approach to in-context link selection optimizes LP performance.

## 6 Conclusion

In this paper, we introduce the Universal Link Predictor, a novel approach designed to be immediately applicable to any non-attributed graph dataset without the necessity of training or finetuning. Recognizing the issue of conflicting connectivity patterns among diverse graph datasets, we innovatively employ ICL to dynamically adjust link representations according to the specific properties of the target graph by conditioning on support links as contextual input. Through extensive experimental evaluations, we have demonstrated the effectiveness of our method. Notably, our Universal Link Predictor excels in its ability to adapt seamlessly to new, unseen graphs, surpassing traditional models that require explicit training. This significant advancement presents a promising direction for future research and applications in the field of LP.

**Limitations** Universal Link Predictor achieves strong link prediction performance on unseen graphs without training. However, it has two main limitations. First, it operates only on non-attributed graphs, leveraging graph structure but ignoring node features. While prior studies reveal that graph structural signals often dominate in the link prediction task, incorporating node attributes can further improve performance. Second, the method depends on SEAL as the structural encoder, which requires costly subgraph extraction and limits scalability. Although SEAL's latent representations are effective for in-context learning, integrating more efficient structural encoders is a promising direction for future work.

## 7 Acknowledgements

We would like to thank the anonymous reviewers for their insightful comments and helpful discussions. This work was supported in part by the National Science Foundation, Grant ITE-2333795.

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

# A    Experimental details

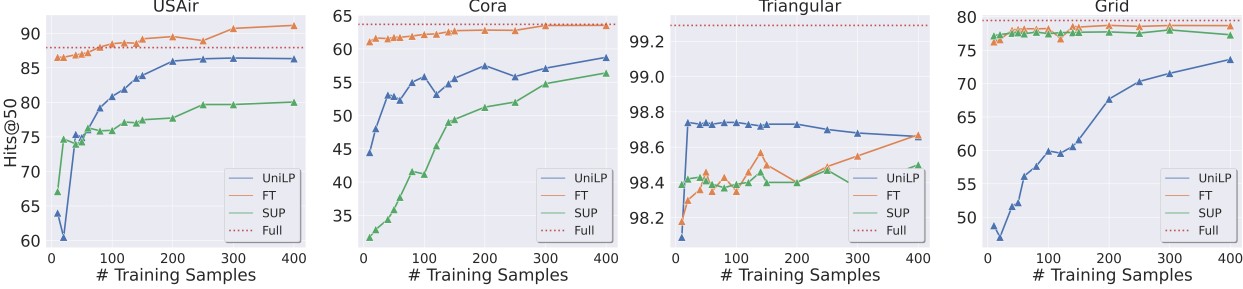

Figure 7: Performance of UniLP with varying quantities of in-context links on the rest of graph datasets.

## A.1 Pretraining the Models

We pretrain UniLP on the datasets listed in Table 1, employing an approach by sampling an equal number of non-connected node pairs $(V \times V \setminus E^o)$ as negative samples to match the count of observed links $(|E^o|)$ in each graph. The pretraining follows a standard binary classification framework.

For predicting each query link, we sample 40 positive and negative links as in-context links $(S^+ \cup S^-)$ from the respective pretrain dataset. This setup ensures variability: different query links from the same dataset or the same query link across training batches may be paired with different in-context links. However, during testing, the set of in-context links for each test dataset remains constant. This training methodology serves multiple purposes: it enhances UniLP's generalization capabilities by exposing it to a broad range of in-context links and optimizes GPU memory usage by selecting a manageable yet diverse set of in-context links during pretraining.

The pretraining phase incorporates an early stopping criterion based on performance across a merged validation set, which comprises 200 links from the validation set of each test dataset. This approach will stop UniLP's optimization once it reaches optimal performance on this merged validation set, ensuring efficiency and preventing overfitting.

## A.2 Software and hardware details

We implement UniLP in Pytorch Geometric framework (Fey & Lenssen, 2019). We conduct our experiments on a Linux system equipped with an NVIDIA A100 GPU with 80GB of memory.

# B Supplementary experiments

## B.1 Synthetic graphs

Table 4: Link prediction results on synthetic Triangular/Grid lattice graphs evaluated by Hits@50. The format is average score ± standard deviation. The top three models are colored by **First**, **Second**, **Third**.

|  | Triangular | Grid |
|---|---|---|
| Heuristics | | |
| **CN** | $73.58_{\pm 0.81}$ | $0.00_{\pm 0.00}$ |
| **AA** | $73.58_{\pm 0.81}$ | $0.00_{\pm 0.00}$ |
| **RA** | $73.58_{\pm 0.81}$ | $0.00_{\pm 0.00}$ |
| **PA** | $0.00_{\pm 0.00}$ | $0.00_{\pm 0.00}$ |
| **SP** | $97.91_{\pm 0.63}$ | $\mathbf{86.04_{\pm 1.11}}$ |
| **Katz** | $90.08_{\pm 0.67}$ | $56.79_{\pm 0.99}$ |
| SEAL | | |
| **Supervised** | $\mathbf{99.29_{\pm 0.28}}$ | $\mathbf{79.45_{\pm 1.09}}$ |
| **Pretrained Only** | $98.11_{\pm 0.84}$ | $61.48_{\pm 0.57}$ |
| **Pretrain & Finetune** | $\mathbf{98.35_{\pm 0.57}}$ | $\mathbf{78.24_{\pm 0.79}}$ |
| Ours | | |
| **UniLP** | $\mathbf{98.73_{\pm 0.49}}$ | $77.39_{\pm 1.38}$ |

We deployed our pretrained UniLP on the synthetic graph from Figure 2a, 2b, with outcomes presented in Table 4. These findings demonstrate that UniLP matches the performance of both models that are fully trained on the entire graph and those that undergo explicit finetuning. This performance underscores the efficacy of UniLP's ICL capability, affirming its ability to dynamically adapt to synthetic graph environments and learn connectivity patterns directly from in-context links without the need for additional training or finetuning.

Table 5: Link prediction results on OGB datasets.

| | OGBL-Collab Hits@50 | OGBL-Citation2 MRR |
|---|---|---|
| Heuristics | | |
| **CN** | 61.37 | 51.47 |
| **AA** | 64.17 | 51.89 |
| **RA** | 64.00 | 51.98 |
| Pretrain Only | | |
| **SEAL** | 45.44 | 80.10 |
| **GAE** | 10.29 | 8.02 |
| **ELPH** | 56.70 | 15.03 |
| **NCNC** | 58.32 | 20.76 |
| **MPLP** | 49.33 | 45.57 |
| Pretrain & Finetune | | |
| **SEAL** | 56.12 | 81.35 |
| **GAE** | 18.92 | 9.06 |
| **ELPH** | 58.47 | 20.36 |
| **NCNC** | 60.25 | 24.79 |
| **MPLP** | 55.08 | 47.13 |
| Ours | | |
| **UniLP** | 61.09 | 83.29 |

## B.2 OGB datasets

We also evaluate UniLP on the OGB (Hu et al., 2021), specifically on Collab and Citation2[3]. The results are shown in Table 5. UniLP achieves better performance compared to all the Pretrain (& Finetune) methods on two large-scale data, demonstrating its adaptability to new graphs without training on their connectivity patterns at all.

## B.3 Empirical analysis of conflicting patterns

Theorem 2.2 provides a theoretical analysis of the conflicting connectivity patterns in Grid and Triangular graphs. In practical link prediction tasks, a portion of edges is often missing or yet to form. To test whether the conflicting patterns persist under such conditions, we conduct an empirical analysis by removing edges from the synthetic graphs.

Following the same setup as in Section 5, we randomly remove 30% of the edges from both the Grid and Triangular graphs. We then estimate $p(y = 1|A_2)$ and $p(y = 1|A_3)$ empirically for each graph. The results, presented in Table 6, show that the two graphs continue to exhibit different connectivity patterns, despite their structural similarity. This supports the conclusion that the conflicting patterns identified in theory also hold under more realistic conditions.

Table 6: Empirical analysis for Theorem 2.2 under realistic conditions.

| | Triangular | Grid |
|---|---|---|
| $p(y = 1|A_2)$ | **0.3184**±0.00199 | **0.0000**±0.00000 |
| $p(y = 1|A_3)$ | **0.1656**±0.00067 | **0.2536**±0.00079 |

---

[3]When evaluating SEAL and UniLP on Citation2, we sample the test set to 1% to get an unbiased estiamte of their performance, following (Zhang et al., 2021) at `https://github.com/facebookresearch/SEAL_OGB?tab=readme-ov-file#ogbl-citation2`.

# C  Theoretical analysis

## C.1  More discussions about the connectivity patterns

In the initial definition (Definition 2.1), connectivity patterns are characterized as ordered sequences of events that are satisfied by the features of links. This concept indicates that if two graphs exhibit identical connectivity patterns, an LP model trained on one graph could theoretically be applied to the other without retraining. The rationale behind this is rooted in the LP task's core objective: to prioritize true links over false ones through ranking. Hence, a consistent ranking mechanism across different graphs allows for the same heuristic-based link predictor to be effectively utilized for LP tasks across those graphs.

It might be tempting to equate connectivity patterns directly with graph distributions; however, this is a misconception. Graphs can share identical connectivity patterns yet differ significantly in their underlying distributions. An illustrative example is provided by graphs generated through the Stochastic Block Model (Holland et al., 1983) with distinct parameters, which may still present identical connectivity patterns as long as their intra-block edge probabilities are higher than those between blocks.

Consequently, despite real-world graphs often exhibiting varied underlying distributions—reflected in aspects such as node degrees, graph sizes, and densities—the question of whether a singular, common connectivity pattern exists across diverse graphs remains non-trivial. This inquiry forms the theoretical foundation for our Universal Link Predictor model, challenging us to explore the feasibility of applying one singular link prediction methodology in a world of inherently distinct graph structures.

## C.2  Proof for Theorem 2.2

We first restate the theorem and proceed with the proof:

Define $A_2 = |\pi_2(u, v)| \geq 1$ and $A_3 = |\pi_3(u, v)| \geq 1$ as elements of $\omega$. The connectivity patterns on Grid and Triangular graphs are distinct. Specifically:
(i) On Grid: $\omega = [A_3, A_2]$; (ii) On Triangular: $\omega = [A_2, A_3]$.

*Proof.* In a Grid graph, the probability of a connection given a 2-hop simple path, $p(y = 1|A_2)$, can be expressed as $\frac{p(y=1, A_2)}{p(A_2)}$. The absence of any 2-hop connected node pairs $(u, v) \in E^o$ implies $p(y = 1, A_2) = 0$, leading to $p(y = 1|A_2) = 0$.

Considering the symmetric nature of nodes in a synthetic Grid graph, we select an arbitrary node as an anchor. Identifying nodes with a 3-hop simple path to this anchor reveals that:

$$p(y = 1|A_3) = \frac{p(y = 1, A_3)}{p(A_3)} = \frac{4}{16} = \frac{1}{4}. \tag{7}$$

This calculation confirms the connectivity sequence on Grid as $\omega = [A_3, A_2]$.

Conversely, in a Triangular graph, the probabilities given a 2-hop and a 3-hop simple path are calculated as:

$$p(y = 1|A_2) = \frac{p(y = 1, A_2)}{p(A_2)} = \frac{6}{18} = \frac{1}{3},$$

$$p(y = 1|A_3) = \frac{p(y = 1, A_3)}{p(A_3)} = \frac{6}{36} = \frac{1}{6}.$$

Thus, establishing the connectivity sequence for Triangular as $\omega = [A_2, A_3]$, which is in direct contrast to that of Grid graphs, highlighting the inherent difference in their connectivity patterns. $\square$

