# OpenReview forum: "Universal Link Predictor By In-Context Learning on Graphs"
_TMLR — Accepted by TMLR_

### Review · Reviewer_bRwi · 2025-02-25

**Summary Of Contributions:**

This paper presents a link prediction framework that augments the traditional GNN-based methods with in-context embeddings. The paper is motivated by unifying the two lines of link prediction works, non-parametric heuristic link predictors and GNN-based link predictors, and tries to combine them via in-context samples from the target graph. Experiments on a diverse set of graph datasets covering multiple different domains show that UniLP outperforms both types of methods above. Detailed ablation studies are conducted to reveal the inner mechanism of UniLP, showing skill learning capabilities from ICL.

**Audience:**

Yes

**Broader Impact Concerns:**

No concern.

**Claims And Evidence:**

Yes

**Requested Changes:**

* Please clarify how non-parametric heuristic methods are related with GNN-based UniLP. What's the relationship between non-parametric heuristic methods and ICL?
* Please discuss the computation complexity. Do we need more efficient subgraph GNNs to handle the overhead from ICL samples?
* Please explain what unique information can be brought by ICL samples. Why such information cannot be captured by local subgraphs?

**Strengths And Weaknesses:**

Strengths
+ The paper is carefully written and most parts are clear. The proposed method / model is precisely defined and experiments are illustrated with sufficient data in appropriate forms.
+ The idea of integrating ICL in link prediction / graph learning is interesting. It has been observed that simple heuristics methods may perform very well on link prediction tasks, making it an intriguing question how to efficiently combine heuristic methods with more expensive GNNs.
+ The authors have conducted rich experiments to showcase the advantages of UniLP.

Weaknesses
- I am not sure how the proposed method can integrate non-parametric heuristic methods. It seems the major novelty lies in the introduction of ICL positives/negatives, where these ICL samples are encoded via GNNs. The non-parametric heuristics are not part of the model, and the ICL also may not be able to express the simple human-tailored heuristics.
- One main concern of the proposed model is in its computation cost. GNN-based link prediction is already expensive to compute. With the additional ICL samples, it becomes much more expensive. To predict 1 candidate link, we basically need to run GNN through all positive / negative "context" links. The authors may consider adding a discussion on how to make UniLP efficient / scalable. e.g., maybe some subgraph-based GNN models like [1] can reduce computation cost by more efficient sampling.
- For experiments, most benchmark datasets are quite old and of small scale. This makes the gains less convincing. It would be better to include results of modern benchmarks like OGB [2]
- It is not intuitive what are the additional information brought by the ICL examples. Since positives / negatives are randomly sampled, they mostly provide global graph information. However, it is arguable that local graph / neighbor information may dominate in many link prediction tasks. In addition, standard GNNs may already encode much "context" information -- if we understand "context" for a pair of nodes as the surrounding subgraph neighborhood.

-----

[1] Decoupling the Depth and Scope of Graph Neural Networks. In NeurIPS 2021. https://arxiv.org/abs/2201.07858

[2] Open Graph Benchmark. https://ogb.stanford.edu

---

> ### Author Response · Authors · 2025-04-17
>
> Dear Reviewer bRwi,
>
> We thank the reviewer for the constructive feedback. Here is our response to your questions.
>
> ## W1/R1: how UniLP can integrate non-parametric heuristic methods.
> The capability of UniLP's integrating non-parametric heuristic methods lies in our DRNL+ node labeling, as discussed in Section 3.2. Consider Common Neighbor, one of the most representative heuristic methods. If a target link $e=(u,v)$ has a common neighbor node $i$, DRNL+ will assign a node label $(2,0)$ to node $i$, meaning this is a common neighbor conditioning on link $e$. Therefore, the link representation $h_e$ will include the information about the Common Neighbor of link $e$, thus integrating the non-parametric heuristic methods.
>
> While our GNN encoder can capture non-parametric heuristic methods, it cannot adjust itself to the pattern of the new unseen graph. ICL plays a role of enabling UniLP to adapt to new graph based on the in-context links' connectivity pattern.
>
> ## W2/R2: Computation Cost
> We appreciate the reviewer's question about UniLP's computation cost. Compared to typical GNN-based link prediction, UniLP indeed requires an extra computation on the in-context links. However, we can greatly reduce the overhead by caching the link representation of the **in-context links** during inference. To score a set of links from a target graph, we first precompute the in-context links' representation. Because the **in-context links** can remain the same when scoring any **target link** on that graph, we can reuse these precomputed representation during inference. In the end, the only extra computation required by UniLP will be that the attention layer (Section 3.3) fuses the target and in-context links' representation together to get the final representation.
>
> Regarding [1], our UniLP also employs a similar strategy as [1] that UniLP performs multiple rounds of message-passing regardless of the hop of the ego-subgraph, as discussed in Section 3.1. We have added [1] into the paper.
>
> ## W3: OGB datasets
> We thank the reviewer's helpful suggestion. Since UniLP is mainly designed to adapt to new unseen graphs across domains without training, we focus on the domain diversity of graph datasets as discussed in the paper.
>
> To further validate the performance of UniLP, we test it on two OGBL datasets:
>
> | Model | Hits@50 | MRR |
> |---|---|---|
> |  | OGBL-Collab | OGBL-Citation2 |
> | Heuristics |  |  |
> | CN | 61.37 | 51.47 |
> | AA | 64.17 | 51.89 |
> | Pretrain Only |  |  |
> | SEAL | 45.44 | 80.10 |
> | GAE | 10.29 | 8.02 |
> | ELPH | 56.70 | 15.03 |
> | NCNC | 58.32 | 20.76 |
> | MPLP | 49.33 | 45.57 |
> | Pretrain & Finetune |  |  |
> | SEAL | 56.12 | 81.35 |
> | GAE | 18.92 | 9.06 |
> | ELPH | 58.47 | 20.36 |
> | NCNC | 60.25 | 24.79 |
> | MPLP | 55.08 | 47.13 |
> | Ours |  |  |
> | UniLP | 61.09 | 83.29 | |
>
> As the result shows, UniLP perform significantly better than other baselines on both OGBL datasets, in both Pretrain-Only and Pretrain-Finetune settings. Even though UniLP is never trained on the target graphs, it can still dynamically adapt to it on the fly. This demonstrates UniLP's generalizability on large-scale graphs.
>
>
> ## W4.1/R3: what are the additional information brought by the ICL examples?
> We appreciate the reviewer's insightful question. Since different target graph can have different connectivity pattern (as discussed in Section 2), UniLP adapts to any new unseen graph by incorporating how links are getting connected on the graph. Therefore, these ICL examples show how the links should be connected. In Section 5.3, we discuss how the ICL examples can impact the adaptability of UniLP, showing that ICL examples are necessary for UniLP to adapt to new graphs.
>
> ## W4.2/R3: standard GNNs may already encode much "context" information.
> We thank the reviewer's question. Standard GNNs can somehow encode "context" information, but they are **not trained** to utilize such context. In Section 2.1, we examine whether GNNs can spontaneously identify the connectivity pattern by only using the surrounding subgraph neighborhood. However, introducing training signals from other graph can harm the performance of GNNs. This indicates that GNNs cannot adapt by its own. In contrast, UniLP is trained to make a link prediction based on the explicitly sampled in-context links. This makes UniLP capable of adapting to new graph based on context.
>
>
> [1] Decoupling the Depth and Scope of Graph Neural Networks.
>
> [2] Open Graph Benchmark.
>
>
> -----
> Thank the reviewer for the helpful feedback. Please let us know if you have any questions.
>
> Best,
> UniLP authors

---

> > ### Comment · Reviewer_bRwi · 2025-04-29
> > **Reviewer's response**
> >
> > I greatly appreciate the authors' responses and revision to the paper. They help clarify many of my initial questions. I would like to share some additional thoughts:
> >
> > W1: While I agree that labeling trick or distance encoding can help express many parametric heuristics, it is arguable if the paper should make such a claim on "unification". First, can labeling trick provably express all (or most) parametric heuristics? Second and more importantly, labeling trick is not the contribution of this paper. Does UniLP propose any new technique that enables more comprehensive unification than existing GNN+labeling trick?
> >
> > W2: Thanks for the clarification. In general, I agree that caching may help with computation cost reduction.
> >
> > W3: Thanks for adding the OGB results. MRR improvement on citation dataset is quite significant. AA on collab performs better than all GNN models, including UniLP. From my understanding, AA can also be applied to new unseen graphs without training, and is more computationally efficient than GNNs. It may be worth clarifying this point.
> >
> > W4.1: Thanks. I now see the difference between neighbor subgraph and ICL subgraphs. ICL examples are explicitly providing label information, since you always put k positives followed by k negatives. So it's reasonable to see benefits when applying UniLP to new graphs where the training process doesn't see any labels of the new graph.
> >
> > W4.2: It makes sense that ICL provides different "context" than subgraph neighborhood. But my feeling is that such difference lies in the explicit label information due to your ICL construction (see W4.1 response). Then the global vs local question still holds. Since ICL are randomly sampled, they would only provide global information. It would be interesting to see a paragraph discussing different ways of picking ICL links: random sampling or sampling from some local neighborhoods. For example, we may pick ICL positives & negatives from local PPR subgraphs, which was shown to provide good local information in [1]. Maybe PPR ICL links can be better than random ICL links on graphs with diverse connectivity patterns?
> >
> > ------
> >
> > [1] Decoupling the Depth and Scope of Graph Neural Networks

---

> > > ### Author Response · Authors · 2025-04-30
> > > **Response Round 2**
> > >
> > > Dear Reviewer bRwi,
> > >
> > > We really appreciate you for your time and effort in reviewing our paper. We answer the new questions as follow:
> > >
> > > ## W1.1: labeling trick's expressiveness
> > > We thank the reviewer for the insightful question. The labeling trick applied in our paper cannot express all heuristics method. We use term "Universal" referring to that our method can be applied to all graphs even not seen during training. However, it cannot fully capture all heuristics.
> > >
> > > In fact, there exists **no** computationally practical method to provably express all heuristics. Expressing all heuristics can be seen as equivalent to graph isomorphism test, which is likely to be NP-hard. Therefore, the design of GNNs, especially for link prediction task, is to find a tradeoff between computation efficiency and model expressiveness. For example, Common Neighbor has been found as one of the most effective signals/graph substructure for link prediction[1,2,3], even though it is just one heuristic.
> > > Similarly, our labeling trick can express certain set of heuristics, but cannot capture them all.
> > >
> > >
> > > ## W1.2: labeling trick' novelty in our paper
> > > The labeling trick employed in our paper, DRNL+, enhances the expressiveness of the previous DRNL[1]. The original DRNL cannot distinguish neighboring nodes only reachable to one of the target node. As an improvement, we use Distance Encoding[4] to assign a tuple of labels for each node, to differentiate those nodes. We found that this simple trick can improve performance by 3% on average compared to previous DRNL measured by Hits@50. Therefore, we adopt it as our labeling trick.
> > >
> > > [1] Zhang, Muhan, et al. "Labeling trick: A theory of using graph neural networks for multi-node representation learning."
> > >
> > > [2] Dong, Kaiwen, Zhichun Guo, and Nitesh Chawla. "Pure message passing can estimate common neighbor for link prediction."
> > >
> > > [3] Wang, Xiyuan, Haotong Yang, and Muhan Zhang. "Neural common neighbor with completion for link prediction."
> > >
> > > [4] Li, Pan, et al. "Distance encoding: Design provably more powerful neural networks for graph representation learning."
> > >
> > >
> > > ## W3: AA is better than all GNNs.
> > > We thank the reviewer for pointing it out. AA is indeed able to be applied to any new graph with better efficiency compared to GNNs. And it also shows the best performance on Collab. We hypothesize that the underlying distribution of Collab happens to fit the AA's criteria. However, when applying AA to benchmarks in our experiment in Section 5, UniLP, with the highest average rank, is better than AA on average. It suggests that heuristics like AA can perform well on certain sets of benchmarks, but in general UniLP is better when applying to a boarder set of graphs.
> > >
> > > ## W4: global vs local context
> > > We appreciate the reviewer's insightful question. We select the globally randomly sampled ICL because of the training paradigm of common link prediction methods. Typically, GNNs for link prediction are also trained on global samples. Therefore, we follow the same principle that **the connectivity pattern is global rather than local**. To the best of our knowledge, there is no study on whether one target link's connectivity pattern is more likely to follow the global pattern or local neighborhood pattern, especially when two differ.
> > >
> > >
> > > We also believe that the study of selecting ICL can be an interesting furture study. In fact, in Section 5.6, we discuss how the selection of ICL can influence the model performance. For graphs like PB, the selection of ICL has minimal impact on the model performance. However, for Cora, a selection of ICLs with high quality may improve the model performance, **which may suggest that global pattern is not the most effective signals on Cora.** We leave this as a future study.
> > >
> > > -----
> > >
> > > Thank you again for the time and effort. Please let us know if you have any questions.
> > >
> > > Best,
> > >
> > > UniLP authors

---

> > > > ### Comment · Reviewer_bRwi · 2025-06-01
> > > > **Reviewer's response**
> > > >
> > > > I appreciate the authors' follow-up responses. They mostly make sense to me. Please add those discussions to your revision, especially to clarify the scope of the contribution. It is fine that the proposed method does not express all heuristics. But it is important to make it clear in the paper what "universal" refers to and what is new / not new in the labeling trick implemented in the proposed model.
> > > >
> > > > The submission would look good to me given that the discussion in the rebuttal is added in the revision.

---

> > > > > ### Author Response · Authors · 2025-06-01
> > > > > **Response Round 3**
> > > > >
> > > > > Dear Reviewer bRwi,
> > > > >
> > > > > We appreciate your time and effort in reviewing our paper. We have added the discussions above in the revision, highlighted in blue:
> > > > >
> > > > > 1. For discussions about the scope of contribution and why it is "Universal", we added it in Section 3.
> > > > >
> > > > > 2. For discussions about the novelty and expressiveness of "Labelling trick", we added it in Section 3.2.
> > > > >
> > > > > ----------
> > > > > Thank the reviewer for the helpful feedback. Please let us know if you have any questions.
> > > > >
> > > > > Best,
> > > > >
> > > > > UniLP authors

---

### Review · Reviewer_1AU4 · 2025-03-16

**Summary Of Contributions:**

In this submission, the authors propose a new link prediction method based on in-context learning, which makes an attempt to bridge the gap between heuristic approaches and learning-based parametric prediction models.

The proposed work first demonstrates that different graphs may have conflicting connectivity patterns, and adapting models to the graphs with conflicting patterns often leads to performance degradation. The authors solve this problem by in-context learning, predicting the links of target graphs based on the positive and negative examples plugged into the learned model.

Finally, experiments show that the proposed method achieves competitive performance, obtaining top-3 link prediction accuracy in most testing datasets.

**Audience:**

Yes

**Claims And Evidence:**

Yes

**Requested Changes:**

1) The analytic content in section 2 can be more solid and convincing. Pls add more analysis on the relation between the graph nodes' homogeneity and the link prediction's transferability.

2) More analytic experiments and ablation studies should be added for the proposed method.

3) More relevant baselines should be introduced. The differences between the proposed method and the baselines should be provided, in both the related work section and the experimental part.

**Strengths And Weaknesses:**

Strengths:

1) The motivation and the principle of the method, in my opinion, are reasonable. In particular, using ICL to improve the generalization power of prediction models is a promising solution, which has been demonstrated in NLP tasks. I think it has the potential in the link prediction task as well.

2) The organization of the paper is reasonable and clear, and the analytic part in section 2 is interesting. Although having some typos, the writing is clear to some extent.

Weaknesses:

1) In section 2, I think the empirical transferability experiment and the theoretical analysis are highly relevant to the homogeneity and the heterogeneity of graphs, which should be analyzed in depth. For example, in Figure 1, besides showing the change of model performance, the homogeneity scores of different graphs (i.e., the proportion of the edges whose nodes have the same label) should be provided as well. I believe the scores are correlated with the performance changes.

2) For the proposed method, the length of contextual graph sequences, the proportion of positive/negative graphs, and the generation strategies of the contextual graphs are key factors impacting the model performance. However, I did not see detailed analytic experiments or ablation studies on these factors. In fact, I believe that the model performance would be much better if the authors could have analyzed the factors and found better model configurations.

3) Some strong baselines using ICL or SSL are missed, e.g., the references shown below. I wonder if there are any difficulties in comparing the proposed method with them.

[1] Wang, Ping, et al. "Self-supervised learning of contextual embeddings for link prediction in heterogeneous networks." Proceedings of the web conference 2021.

[2] Daza, Daniel, Michael Cochez, and Paul Groth. "Inductive entity representations from text via link prediction." Proceedings of the Web Conference 2021.

[3] Huang, Qian, et al. "Prodigy: Enabling in-context learning over graphs." Advances in Neural Information Processing Systems 36 (2023): 16302-16317.

---

> ### Author Response · Authors · 2025-04-17
>
> Dear Reviewer 1AU4,
>
> We thank the reviewer for the constructive feedback. Here is our response to your questions.
>
>
> ## W1/R1: homophily scores of graphs in Section 2.
> We appreciate the reviewer's insightful suggestion. The reviewer suggests that we can analyze the proportion of the edges whose nodes have the same label for graphs in Section 2. We respectfully assume that the reviewer is referring to homophily scores for those graphs rather than homogeneity.
>
> We also think this can be an interesting analysis comparing homophily scores and the link prediction's transferability. However, the graphs used in Section 2 do not come with the node label data. In fact, on most of graphs, the node label data is often not naturally present. Therefore, in Section 2, we study the link prediction's transferability for a broader set of graphs, even when the node label is not available.
>
> ## W2/R2: length of contextual graph sequences, proportion of positive/negative graphs, and generation strategies of the contextual graphs
> We appreciate the reviewer's insightful question. In the experimental section of our paper, we have discussed how the selection of in-context links can impact the performance of UniLP.
>
> **Length of contextual graph sequences:** In Section 5.4, we conduct an ablation study on the length of in-context links. It shows that the performance of UniLP can consistently improve with more in-context links. This is intuitive because more in-context links can better reflect the connectivity pattern of the target graph.
>
> **proportion of positive/negative graphs:** In Section 5.7, we conduct an ablation study on varying the positive-to-negative ratios of in-context links. It shows that different graph can have different optimal ratio to achieve the best performance. For example, the synthetic GRID graph prefers negative in-context links to learn how negative node pairs are disconnected. However, Facebook prefers a balanced set of positive/negative examples for the optimal performance.
>
> **generation strategies of the contextual graphs:** In Section 5.6, we discuss how the selection of in-context links can influence the model performance. For graphs like PB, the selection of in-context links has minimal impact on the model performance. However, for Cora, a selection of in-context links with high quality may improve the model performance. In addition, we also discuss how randomly generated in-context links can impact the performance in Section 5.3. In general, random context (marked as **UniLP-RandomContext** in Table 3) can degenerate the model performance with varying degree.
>
> In summary, we have conducted comprehensive ablation studies to investigate how in-context links may influence the model. We find that the selection of in-context links can significantly impact the model performance. It would be an interesting future work to study how to compose an optimal set of in-context links for the best performance.
>
> ## W3/R3: Relevant works
> We appreciate the reviewer's suggestion on the relevant works. We have added the discussion of the relevant works in Section 4 of the revised manuscript, hightlighted in blue.
>
> However, we find it difficult to directly compare UniLP to these relevant works due to the problem setting. More specifically, [1] focuses on the link prediction on the heterogeneous networks and fails to transfer to unseen new graphs. [2] requires that the graphs come with text descriptions associated with nodes, where UniLP can work on any graphs without node attributes. For [3], we have initially included the related work discussion in Section 4. The pretrain and testing graphs of Prodigy[3] has some overlap because the pretrain data MAG is a superset of the testing graphs Arxiv. In contrast, UniLP is pretrained on graphs never used in the testing.
>
>
> [1] Wang, Ping, et al. "Self-supervised learning of contextual embeddings for link prediction in heterogeneous networks." Proceedings of the web conference 2021.
>
> [2] Daza, Daniel, Michael Cochez, and Paul Groth. "Inductive entity representations from text via link prediction." Proceedings of the Web Conference 2021.
>
> [3] Huang, Qian, et al. "Prodigy: Enabling in-context learning over graphs." Advances in Neural Information Processing Systems 36 (2023): 16302-16317.
>
> -----
> Thank the reviewer for the helpful feedback. Please let us know if you have any questions.
>
> Best,
> UniLP authors

---

> > ### Comment · Reviewer_1AU4 · 2025-05-16
> > **More comments**
> >
> > Thanks for your responses. However, my concerns are not fully resolved.
> >
> > 1. For my first comment, I still believe that analyzing the correlation between graphs' homophily scores and the link prediction's transferability can provide more useful insights and make the work more valuable. Besides the graphs used in the current submission, the authors can apply some graphs with node labels and construct link prediction tasks for them (e.g., masking some node pairwise relations for prediction) in this analytic experiment.
> >
> > 2. For my third comment, although the baselines are different from the proposed method in experimental settings, some of them are still comparable, and moreover, the comparisons are meaningful. For example, by comparing with [2], we can see whether the proposed method can approach the competitor using textual side information. I don't expect the proposed method to consistently work better than the baselines using side information. Instead, what I really care about is whether the experiment is comprehensive and solid.
> >
> > In summary, I ask for one more round of review for this work and hope to see more analytic experiments in the revised paper.

---

> > > ### Author Response · Authors · 2025-05-18
> > >
> > > Dear reviewer 1AU4,
> > >
> > > We really appreciate you for your time and effort in reviewing our paper. We answer the new questions as follow:
> > >
> > > 1. To further analyze the correlation between graphs' homophily scores and the link prediction's transferability, we reproduced the experiment in Section 2.1, including both homophilous and heterophilous graphs. The homophily score is defined as [1].
> > >
> > > | Homophily Scores | 0.825 | 0.718 | 0.792 | 0.247 | 0.217 | 0.215 |
> > > |---|---|---|---|---|---|---|
> > > | Performance Change in Hits@50 | Cora | Citeseer | Pubmed | Chameleon | Squirrel | Actor |
> > > | Add Cora | 0 | +1.9 | -1.4 | -3.0** | -1.5 | -0.3 |
> > > | Add Citeseer | -2.2 | 0 | +0.8 | -0.8 | -0.7 | -1.2 |
> > > | Add Pubmed | +2.1 | -0.6 | 0 | +1.2 | +0.8 | -1.5 |
> > > | Add Chameleon | -5.4** | -3.2** | -3.0** | 0 | +1.2 | -0.4 |
> > > | Add Squirrel | -4.6** | -4.8** | -1.2 | +0.8 | 0 | -0.8 |
> > > | Add Actor | -1.3 | -0.8 | -0.3 | -0.2 | -0.6 | 0 |
> > >
> > >
> > > Similarly, the results above also show that adding an extra graph into training generally results in performance degradation, suggesting conflicts between the connectivity patterns across graphs.
> > >
> > > Moreover, it reveals an interesting trend between homophilous and heterophilous graphs. After adding heterophilous graphs as additional training signals, the model performance on homophilous graphs drops significantly. In contrast, adding homophilous graphs brings statistically minor impact to the model performance on the heterophilous graphs. This may suggest that heterophilous graphs have a more severe influence on what the model would learn during training for link prediction tasks.
> > >
> > > 2. We thank the reviewer's suggestion. To validate if textual side information can improve the model performance, we adapted the method proposed in [2] to our experiment setting.
> > >
> > > One of the main contributions in [2] is to apply a Pretrained Language Model like BERT to encode the raw text associated with nodes in graph. Therefore, it aligns the feature space of node attributes across graphs and leads to a potential transfer learning. However, [2] mainly discusses link prediction as a knowledge graph (KG) completion task, where there are various relation types. In our study, the experiment setting focuses on whether the links (relation) exist, rather than the type of links. Therefore, to utilize the textual side information, we follow the application of BERT encoding raw text data in [2], but not their specific designs for KG (TransE, ComplEx, DistMult) in the following experiment.
> > >
> > > We include Cora, CitationV8[3], and GoodReads[3] as the graph benchmark which comes with raw text data. We use BERT-base as the text encoder. We follow the same experimental setting as of Section 5 in the paper and use SEAL and GAE as the baselines, pretrained on Citeseer, Pubmed, OGBN-Arxiv, and OGBN-Products. During evaluation, we downsample the testing edges in CitationV8 and GoodReads to 30000 so that they can be scored in a reasonable amount of time. The results are shown below:
> > >
> > >
> > >
> > > | Model (Hits@50) | Cora | CitationV8 | GoodReads |
> > > |---|---|---|---|
> > > | Heuristics |  |  | |
> > > | CN | 33.85 |  57.26 | 61.39|
> > > | AA | 33.85 | 58.33 |  63.84|
> > > | Pretrain Only |  |  | |
> > > | SEAL | 56.21 | - | -|
> > > | GAE | 24.22 | - | -|
> > > | SEAL (text) | 58.21 | 60.21 | 65.19|
> > > | GAE (text) | 46.38 | 30.25 | 55.12|
> > > | Pretrain & Finetune |  |  |
> > > | SEAL | 62.19 | - | -|
> > > | GAE  | 25.31 | - | -|
> > > | SEAL (text) | 60.50 | 65.12 | 66.28|
> > > | GAE (text) | 52.13 | 45.88 | 58.84|
> > > | Ours |  |  |
> > > | UniLP | 57.50 | 64.79 | 58.28 |
> > >
> > > By adding textual data as side information, there is a significant performance improvement for previous baselines like SEAL and GAE on Cora. Specifically, the performance of GAE almost doubles with additional text features. This suggests that while GAE is not powerful at capturing graph structural features, the complement node features can still boost its performance on link prediction task.
> > >
> > > On datasets like CitationV8 and GoodReads, our proposed UniLP, without node attributes as input, can perform as well as the baselines with textual data. The benefits of textual data suggest that we may further improve UniLP's performance when textual data is available. However, it is still a challenge to design a single model handling graphs both with and without node attributes. We leave this as future study.
> > >
> > >
> > >
> > >
> > > [1] Pei, H., Wei, B., Chang, K. C., Lei, Y., and Yang, B. Geom-gcn: Geometric graph convolutional
> > > networks
> > >
> > > [2] Daza, Daniel, Michael Cochez, and Paul Groth. "Inductive entity representations from text via link prediction." Proceedings of the Web Conference 2021.
> > >
> > > [3] Yan, Hao, et al. "A comprehensive study on text-attributed graphs: Benchmarking and rethinking."
> > >
> > > ----
> > >
> > > Thank you again for the time and effort. Please let us know if you have any questions.
> > >
> > > Best,
> > >
> > > UniLP authors

---

> > > > ### Comment · Reviewer_1AU4 · 2025-05-19
> > > > **Feedback to the new results**
> > > >
> > > > Thanks for your efforts. The new results are reasonable and make this work promising. Please add them to the final version.

---

> > > > > ### Author Response · Authors · 2025-05-20
> > > > >
> > > > > Thanks again for the reviewer's insightful feedback. We will add these new discussions to the final version.
> > > > >
> > > > > -----
> > > > >
> > > > > Best,
> > > > >
> > > > > UniLP authors

---

### Review · Reviewer_ZQjd · 2025-04-10

**Summary Of Contributions:**

This paper presents UniLP, a framework for universal link prediction on non-attributed graphs via in-context learning (ICL). Unlike conventional heuristic and parametric approaches, UniLP is designed to generalize across diverse graph datasets without the need for graph-specific training or fine-tuning. The model leverages contextually sampled support links from a target graph to dynamically adjust its link prediction behavior using an attention-based mechanism, allowing it to align with the graph’s unique connectivity pattern.

**Audience:**

Yes

**Broader Impact Concerns:**

N/A.

**Claims And Evidence:**

Yes

**Requested Changes:**

See Weakness

**Strengths And Weaknesses:**

**Strengths**

1.	The paper introduces a principled integration of ICL into graph machine learning for link prediction, drawing a compelling parallel to recent advances in NLP where LLMs leverage ICL for generalization.

2.	The proposed model, UniLP, outperforms pretrained link prediction frameworks across various datasets and settings, demonstrating generalization ability without task-specific fine-tuning.

3.	The study addresses the important and timely problem of generalizing link prediction models to entirely unseen graphs, a direction of significant practical relevance and theoretical interest.

**Weaknesses**

1.	The method samples a large number of in-context links (up to 400 per query), which may incur substantial computational overhead, especially on large-scale graphs. I strongly recommend that the authors report the pretraining time and evaluation time of UniLP in comparison with standard baselines, including GNN-based models such as SEAL and heuristic models like RA.

2.	The model’s performance appears sensitive to the quality and composition of in-context links. Evaluating this sensitivity by sampling in-context links with different random seeds could provide a clearer understanding of its robustness and the variance in performance due to context selection.

3.	In the theoretical section, the proof for Theorem 2.2 requires revision. First, the proof is located in Appendix C.2, not in the section labeled “2.2” as currently stated. Second, the proof assumes that all edges—except for the target edge—are not missing, which deviates from typical link prediction scenarios where a subset of edges are randomly removed. I recommend that the authors provide an additional proof or discussion under a more realistic setting, such as random edge deletion at a fixed proportion.

4.	The paper omits several important baselines that combine heuristic and learning-based methods, which are relevant to the goal of improving generalization. Notably: [1] propose a method that incorporates topological features using persistent homology, which can be seen as a form of structural heuristic combined with learning. [2] present a general GNN framework that unifies traditional algorithms with neural message passing, which could potentially generalize to unseen graphs as well.

5. Currently, the pretraining and test graphs used in the experiments are from the same domain or category (e.g., social networks, citation networks, or collaboration networks). It would be significantly more compelling to evaluate UniLP under cross-domain generalization, such as pretraining on social networks and testing on biological graphs. This would test the model’s ability to handle more diverse structural and statistical properties, and better reflect real-world deployment scenarios.

6. there is a line of work on inductive relation prediction [3, 2], where models are trained on some graphs and validated on totally unseen graphs, often within the same domain (e.g., knowledge graphs). This setting closely resembles the problem setup in this paper. The authors are encouraged to discuss the similarities and differences between the proposed setting and inductive relation prediction settings, especially in terms of generalization objectives, assumptions, and modeling strategies.

[1] Yan et al. "Link prediction with persistent homology: An interactive view." ICML 2021.

[2] Zhu et al. "Neural bellman-ford networks: A general graph neural network framework for link prediction." NeurIPS 2021.

[3] Teru et al. "Inductive relation prediction by subgraph reasoning." ICML 2020.

---

> ### Author Response · Authors · 2025-04-17
> **Response [1/2]**
>
> Dear Reviewer ZQjd,
>
> We thank the reviewer for the constructive feedback. Here is our response to your questions.
>
>
> ## W1: Computation time comparison.
> We thank the reviewer's helpful suggestion. To examine UniLP's computational time, we compare its walltime against others on testing datasets. Notably, UniLP is one model serving all. Its adaptation to new data only requires inference. Conversely, the computation time of baselines includes both training and inference.
> |Time (s)|C.ele|USAir|Cora|NS|PB|#hyperparameters|
> |-|-|-|-|-|-|-|
> |SEAL|462|361|638|471|3425|x144|
> |ELPH|14|17|18|11|136|x6912|
> |NCNC|9|11|19|7|88|x1.6E11|
> |MPLP|15|21|24|14|95|x2.9E5|
> |UniLP|10|12|25|11|123|x1|
>
> It shows that UniLP has comparable inference time to training and inferencing the standard baselines. Moreover, the training walltime is significantly underestimated considering extensive hyperparameter (**#hyperparameters**) tuning for optimal model. For example, ELPH and NCNC have 6 and 8 different learning rates, not to mention other hyperparameters. Since total number of hyperparameters for baselines can exceed 100 settings, UniLP can show 100x speed advantage when considering the repetitive training time of standard baselines.
>
> For the pretrain stage, UniLP has undergone an extensive training on the pretrain datasets. It takes ~18 hours for UniLP to converge for pretrain. Note that this is a one-time computation expense, and UniLP can be applied to any new unseen graph after the pretrain.
>
> ## W2: Selection of in-context links
> We appreciate the reviewer's insightful feedback. In Section 5.7, we examine how the model performance is sensitive to the selection of in-context links. We sample in-context links with different random seeds and evaluate the variance of model performance. As the result suggests, the model shows varying degree of sensitivity to the selection of in-context links. For example, on graphs like PB, the selection of in-context links has minimal impact on the model performance. However, for Cora, the quality of in-context links has huge impact on the model performance. It is an interesting future work to find the optimal set of in-context links.
>
> Beyond the sampling different set of in-context links, we also examine how (1) the size of in-context links (Section 5.4), (2) the positive-to-negative ratios (Section 5.7), (3) the corruption of in-context links (Section 5.3) can lead to the variance of model performance.
>
> ## W3: Theoretical conclusion under realistic setting
> We really appreciate the reviewer's constructive feedback on our theoretical analysis. To validate the theoretical conclusion under a more realistic setting, we further conduct an empirical analysis for the conflicting patterns in the two structurally similar graphs in Appendix B.2 (highlighted in blue).
>
> In the experiment, we randomly drop 30\% of edges in the graph to conform a more realistic setting. Then, we empirically estimate the connectivity patterns on the Grid and Triangular graphs. The results show that the two graphs continue to exhibit different connectivity patterns, despite their structural similarity. This supports the conclusion that the conflicting patterns identified in theory also hold under more realistic conditions.
>
> ## W4: Other baselines
> We thank the reviewer for the feedback on relevant baselines. In our experiment, we select baselines that are representative homogeneous link prediction models. We have tried to implement TLC-GNN[1] and NBFNet[2] under our Pretrain (& Finetune) experimental setting. However, both methods are not designed to be pretrained on a diverse set of graphs. For example, TLC-GNN requires a precomputation for the persistence image of edges to be scored. However, it takes >24 hours to preprocess our pretrain graphs like Physics, Twitch, or Github. Similarly, the current implementation of NBFNet is optimized for knowledge graph completion tasks, but it struggles to scale on our pretrain grpah datasets. Therefore, we cannot apply TLC-GNN and NBFNet under our Pretrain (& Finetune) experimental setting.
>
> To evaluate their performance on our benchmark, we switch back to a typical link prediction setting, where the model is **individually** trained on the target dataset and then evaluated. The results are shown below:
>
> |  | C.ele | USAir | PB | NS | CS | Facebook |
> |---|---|---|---|---|---|---|
> | TLC-GNN | 49.78±2.99 | 70.70±3.17 | 24.22±1.70 | 74.52±2.18 | 31.64±7.03 | >24h |
> | NBFNet | 59.03±3.70 | 79.87±2.58 | 48.57±2.35 | 83.26±2.12 | 61.41±4.84 | 60.67±2.63 |
> | UniLP | 65.20±4.40 | 85.98±2.00 | 48.14±2.99 | 89.09±2.05 | 64.59±2.65 | 65.49±2.05 |
>
>
> As the results show, UniLP performs significantly better than the other two baselines, even without being explicitly trained on the target graphs. This demonstrates the capability of UniLP to transfer knowledge to new unseen graphs by adapting on the fly.

---

> > ### Author Response · Authors · 2025-04-17
> > **Response [2/2]**
> >
> > ## W5: UniLP under cross-domain generalization
> > We appreciate the reviewer's insightful suggestion. Since UniLP has already been pretrained on Biology, Transport, Web, Collaboration, Citation, and Social networks, we need to evaluate UniLP on a different domain to test its cross-domain generalizability. We select Photo[4], a **product** network, as our new testing benchmark and report UniLP's performance against pretrain&finetuned baselines:
> >
> > | Model | Photo (Product) |
> > |---|---|
> > | Pretrain & Finetune |  |
> > | SEAL | 37.74±3.15	 |
> > | GAE | 6.54±0.83 |
> > | ELPH | 35.15±3.86 |
> > | NCNC | 37.25±2.59 |
> > | MPLP | 40.19±0.52 |
> > | Ours |  |
> > | UniLP | 38.16±3.95 |
> >
> > As the result shows, UniLP has comparable performance as other SOTA link prediction models explicitly finetuned on the new domain, even though UniLP has never been pretrained on such a domain. This suggests that UniLP has the capability to generalize cross domains by adapting to the target graph during inference.
> >
> > ## W6: Relevant works
> > We thank the reviewer's constructive feedback. We have added the discussion of the relevant works in Section 4 of the revised manuscript, hightlighted in blue. NBFNet[2] proposes a general framework of learning paths for link prediction on knowledge graph. [3] is designed for knowledge graph completion task by applying the node labeling trick introduced in [5]. Similarly, our UniLP also employs a node labeling trick (Section 3.2) with better expressiveness. While [2,3] can generalize to knowledge graph with new entities, they can only be applied to knowledge graph with the same relation types and domains. In contrast, our UniLP is designed to adapt to new graphs that are not seen during pretrain.
> >
> >
> > [1] Yan et al. "Link prediction with persistent homology: An interactive view." ICML 2021.
> >
> > [2] Zhu et al. "Neural bellman-ford networks: A general graph neural network framework for link prediction." NeurIPS 2021.
> >
> > [3] Teru et al. "Inductive relation prediction by subgraph reasoning." ICML 2020.
> >
> > [4] Shchur, O., Mumme, M., Bojchevski, A., & Günnemann, S. (2018). Pitfalls of graph neural network evaluation
> >
> > [5] Zhang, Muhan, et al. "Labeling trick: A theory of using graph neural networks for multi-node representation learning." Neurips 2021.
> >
> >
> > -----
> > Thank the reviewer for the helpful feedback. Please let us know if you have any questions.
> >
> > Best,
> > UniLP authors

---

### Decision · Action_Editor_CLrB · 2025-06-19

**Recommendation:** Accept with minor revision

**Additional Comments:**

Please include results and discussions in the rebuttal into the manuscript.

**Audience:**

Yes

**Audience Explanation:**

Link prediction in graph is an important task and the method will be applicable to many domains.

**Claims And Evidence:**

Yes

**Claims Explanation:**

This paper introduces UniLP, a framework for universal link prediction on non-attributed graphs using in-context learning (ICL). Unlike conventional heuristic or parametric approaches, UniLP is designed to generalize across diverse graph datasets without graph-specific training or fine-tuning. The model dynamically adjusts its link prediction behavior by leveraging contextually sampled support links from a target graph. This is achieved through an attention-based mechanism, allowing UniLP to align seamlessly with each graph’s unique connectivity pattern. Experiments as well as additional results provided during rebuttal have shown the superior power of the proposed method.